# Development of a potent monoclonal antibody for treatment of human metapneumovirus infections

Evelyn D. Harris[1,6], Morgan McGovern [1,6], Sara Pernikoff [1], Ren Ikeda [1], Lea Kipnis[1], William Hannon[2], Elizabeth B. Sobolik[3], Matthew Gray [1], Alexander L. Greninger [3], Sijia He[4], Chen-Ni Chin [4], Tong-Ming Fu [4], Marie Pancera [1] ✉ & Jim Boonyaratanakornkit [1,5] ✉

Human metapneumovirus (HMPV) is a major cause of respiratory infections, particularly among vulnerable populations, yet effective therapeutics remain unavailable. Monoclonal antibodies (mAbs) offer a promising approach for treatment and prevention. We describe the discovery and characterization of 4F11, a highly potent neutralizing mAb with in vitro and in vivo efficacy against HMPV. Using cryo-electron microscopy, we define a unique mechanism of binding employed by 4F11. 4F11 targets an epitope located at the apex of the prefusion F protein (site Ø) with a 1:1 stoichiometry of Fab to trimer, distinct from the 3:1 stoichiometry observed with other HMPV site Ø antibodies. Unlike other site Ø antibodies which penetrate the glycan shield between Asn57 and Asn172, 4F11 binds vertically and directly interacts with the Asn172 glycan. In vitro, 4F11 displays high neutralization potency across diverse HMPV strains. It also shows low propensity for resistance development, with only a single escape mutation (K179E) identified, a mutation not found in any published HMPV sequence to date. Viruses rescued with the K179E escape mutation have significantly decreased fitness in vitro. In male hamsters, 4F11 significantly reduces viral loads in the lungs and nasal turbinates. These findings highlight 4F11 as a promising candidate for therapeutic development.

Human metapneumovirus (HMPV) is a ubiquitous respiratory virus. The global burden of infection is large, with an estimated 14.2 million acute lower respiratory infections annually associated with HMPV in children under 5 years of age[1]. Disease burden is also high for adults aged 60 years or older[2,3]. While several vaccines for older adults to prevent infection caused by respiratory syncytial virus (RSV), a virus related to HMPV, were approved by the Food and Drug Administration in 2023[4–7], protective vaccines for HMPV are not yet clinically available. Even if protective vaccines existed for HMPV, vaccination of highly

immunocompromised individuals rarely achieves protective immunity. Additionally, vaccination prior to immune-ablative therapies is often ineffective or wanes quickly, failing to maintain durable protection[8–10]. HMPV represents a serious threat to immunocompromised patients, particularly hematopoietic stem cell transplant recipients who have mortality rates as high as 43%[11–13]. In adults with other risk factors, the burden of disease from HMPV is comparable to RSV[14,15].

The administration of neutralizing monoclonal antibodies (mAbs) provides an effective alternative to vaccination for prophylaxis and is a

[1]Vaccine and Infectious Disease Division, Fred Hutchinson Cancer Center, Seattle, WA, USA. [2]Human Biology Division, Fred Hutchinson Cancer Center, Seattle, WA, USA. [3]Department of Laboratory Medicine and Pathology, University of Washington, Seattle, WA, USA. [4]IgM Biosciences, Doylestown, PA, USA. [5]Department of Medicine, University of Washington, Seattle, WA, USA. [6]These authors contributed equally: Evelyn D. Harris, Morgan McGovern. ✉ e-mail: mpancera@fredhutch.org; jboonyar@fredhutch.org

relatively safe therapeutic modality. Despite advances in mAb development for the prevention of RSV[16,17] and SARS-CoV-2[18,19], no mAbs are currently available in the clinic that can prevent or treat HMPV infection. The efficacy of mAbs for the treatment of respiratory viral infections is generally lower compared to prophylaxis, because mAbs in the therapeutic setting are administered after the virus has already established a foothold in the host[20–26]. As a result, most mAbs targeting respiratory viruses in development and in clinical use are intended for prophylaxis rather than therapy. To achieve efficacy as a therapeutic modality for HMPV, we sought to identify mAbs with the following three characteristics: (1) high potency; (2) the ability to neutralize multiple strains of HMPV; and (3) low susceptibility to resistance development. Ideally, a single mAb would neutralize all four major subtypes of HMPV (A1, A2, B1, and B2), which cause disease and circulate globally[27,28].

HMPV expresses a class I fusion (F) protein, an essential surface glycoprotein, specialized in mediating fusion between viral and host cell membranes during viral entry. The F protein transitions between a metastable prefusion (preF) conformation and a stable postfusion (postF) conformation[29,30]. Since preF is the major conformation on infectious virions, antibodies to preF tend to be the most potent at neutralizing virus[31–34]. The rapid demise of several mAb candidates authorized for use during the COVID-19 pandemic has also underscored the formidable force of viral evolution and the need to determine how and if a virus can develop mAb resistance. Unlike SARS-CoV-2, which has a high mutation rate, particularly in the spike protein targeted by therapeutic mAbs[35,36], HMPV evolves more slowly, and the F protein has shown minimal antigenic drift over time[37,38]. This slower rate of antigenic drift in HMPV offers a strategic advantage when designing neutralizing mAbs, as the risk of rapid resistance development is comparatively lower. In the present study, we leveraged recombinant F protein stabilized in the preF conformation to identify and clone a highly potent neutralizing mAb, 4F11, from human memory B cells. 4F11 targets a novel, apical epitope that is present only in the preF conformation. 4F11 neutralizes diverse strains of HMPV, has low susceptibility to the development of viral escape, and has therapeutic efficacy in a hamster challenge model. These features together indicate that 4F11 has potential as a therapeutic modality in humans, which could be beneficial to at-risk populations who are at a significant immunological disadvantage in the setting of an infection with HMPV.

## Results

### Identification of a highly potent neutralizing antibody to HMPV
HMPV is divided into two major subtypes, A and B. Although the majority of HMPV-neutralizing B cells (approximately 75%) produce antibodies that neutralize both subtypes A and B[39], we leveraged a "bait and switch" strategy to further enrich for B cells that could neutralize both strains. This strategy is based on the rationale that B cells capable of binding to one strain, while neutralizing another strain, are more likely to cross-neutralize both strains (Fig. 1a). We used fluorescent tetrameric probes to identify individual B cells able to bind the recombinant F protein of HMPV subtype B2, which was stabilized in the preF conformation by introduction of a GCN4 trimerization domain at the C-terminus[40]. A total of 600 million human splenocytes from two donors and 320 million peripheral blood mononuclear cells from another two donors were mixed with tetramers of HMPV B2 preF conjugated to allophycocyanin (APC) and control tetramers of His tag peptide conjugated to APC/Dylight755. This was followed by magnetic enrichment with anti-APC microbeads. Since virtually all adults have been exposed to HMPV[41], it was not necessary to pre-screen donors for seropositivity. A total of 1140 isotype-switched (IgD-) B cells that bound HMPV B2 preF tetramers, but not control tetramers of RSV preF, were individually sorted into wells containing CD40L/IL2/IL21-producing 3T3 feeder cells and then incubated for 13 days to stimulate antibody secretion into culture supernatants (Fig. 1b). To identify wells

containing candidate B cells expressing cross-neutralizing antibodies, culture supernatants from the individually sorted HMPV B2 preF-binding B cells were mixed with live HMPV subtype A2 virus and screened for their ability to reduce plaque formation. Six out of the 1140 HMPV-binding B cells produced antibodies that could neutralize HMPV A2. From these 6 cells, the expressed heavy and light chain genes of 3 B cell clones were sequenced successfully, and their antibodies were named 4F11, 4E11, and 3B5. Based on sequencing analysis, 4E11 and 4F11 were derived from B cells expressing the IgG isotype. Our sequencing analysis was unable to determine the isotype for 3B5. Since palivizumab, nirsevimab, and other mAbs currently approved or authorized for respiratory viral infections utilize an IgG1 constant region, the B cell receptors from the 3 B cell clones were cloned and produced for further study as IgG1. All 3 mAbs had neutralizing activity against HMPV A2 strain CAN97-83, with 4F11 being the most potent in a 50% plaque reduction neutralization test (PRNT$_{50}$) at 5.1 ng/mL (Fig. 1c). For comparison, the RSV and HMPV cross-neutralizing mAb, MxR, was less potent with a PRNT$_{50}$ of 229.4 ng/mL. To assess the neutralization capacity of the 4F11 antibody against different HMPV strains, we used a panel of clinical isolates representing all four subtypes of HMPV (A1, A2, B1, and B2). The 4F11 antibody demonstrated neutralizing activity against all tested subtypes (PRNT$_{50}$ range 0.97 ng/mL for subtype A1 to 10.7 ng/mL for A2 strain IA27-2004) (Fig. 1d). These results highlight 4F11's neutralizing activity across diverse HMPV strains.

### Molecular structure of 4F11 in complex with HMPV F
To map the antigenic binding site of our most potent mAb, 4F11, we performed cross-competition binding experiments with other previously mapped mAbs using the HMPV preF that was partially stabilized by a GCN4 trimerization domain. The 4F11 epitope did not appear to overlap with the epitope of the MxR mAb[42], which we previously isolated and mapped to the equator (site III) of HMPV preF (Figs. 2a and S1a). However, 4F11 did compete, albeit only partially, with the ADI-61026 and SAN32-2 mAbs[39,43] (Figs. 2a and S1b), both of which represent a rare class of antibodies that can penetrate the dense glycan shield to bind at the apex (site Ø) of HMPV preF.

To determine the molecular interactions that mediate how 4F11 binds to HMPV preF, cryoEM analysis was used. We first attempted to complex 4F11 with the HMPV preF trimer that was partially stabilized by a GCN4 trimerization domain, but the trimer dissociated into monomers that were bound to 4F11 Fab. In an attempt to obtain a structure with an intact HMPV preF trimer, we turned to alternative versions of prefusion-stabilized HMPV F trimers. We first turned to DS-CavEs2[44], which includes a Pro185 mutation at site Ø, introduced to confer local rigidity and disfavor helix formation required during the transition to the postfusion conformation. However, 4F11 was unable to bind to this stabilized trimer, whereas the site III mAb MxR retained binding to DS-CavEs2 (Fig. S1c, d). Since our competition experiments indicated that 4F11 bound at or near the apex of GCN4-stabilized HMPV preF (Figs. 2a and S1b), we hypothesized that the Pro185 mutation is part of the 4F11 epitope and could disrupt binding of 4F11. Indeed, reverting proline 185 back to an alanine rescued binding of 4F11 (Fig. S1c). We then attempted to complex 4F11 Fab with the reverted DS-CavEs2 P185A trimer at a 1.2 :1 molar ratio of Fab to trimer, but we still only observed monomers bound to 4F11 Fab, indicating that 4F11 does not stabilize the trimer. In contrast to 4F11, MxR binds across protomers and could help stabilize the trimer. The addition of MxR at a 3:1 molar ratio of Fab: DS-CavEs2 P185A trimer led to a peak in the SEC representing intact trimer bound to three Fabs (Fig. S2). To help stabilize the trimer, MxR Fab was precomplexed with DS-CavEs2 P185A trimer before the addition of 4F11 Fab. After addition of 4F11 to this complex, the peak representing the trimer prebound to three MxR Fabs disappeared, and a new peak appeared that was shifted to the right and exited the SEC at a fraction corresponding to the size of the F

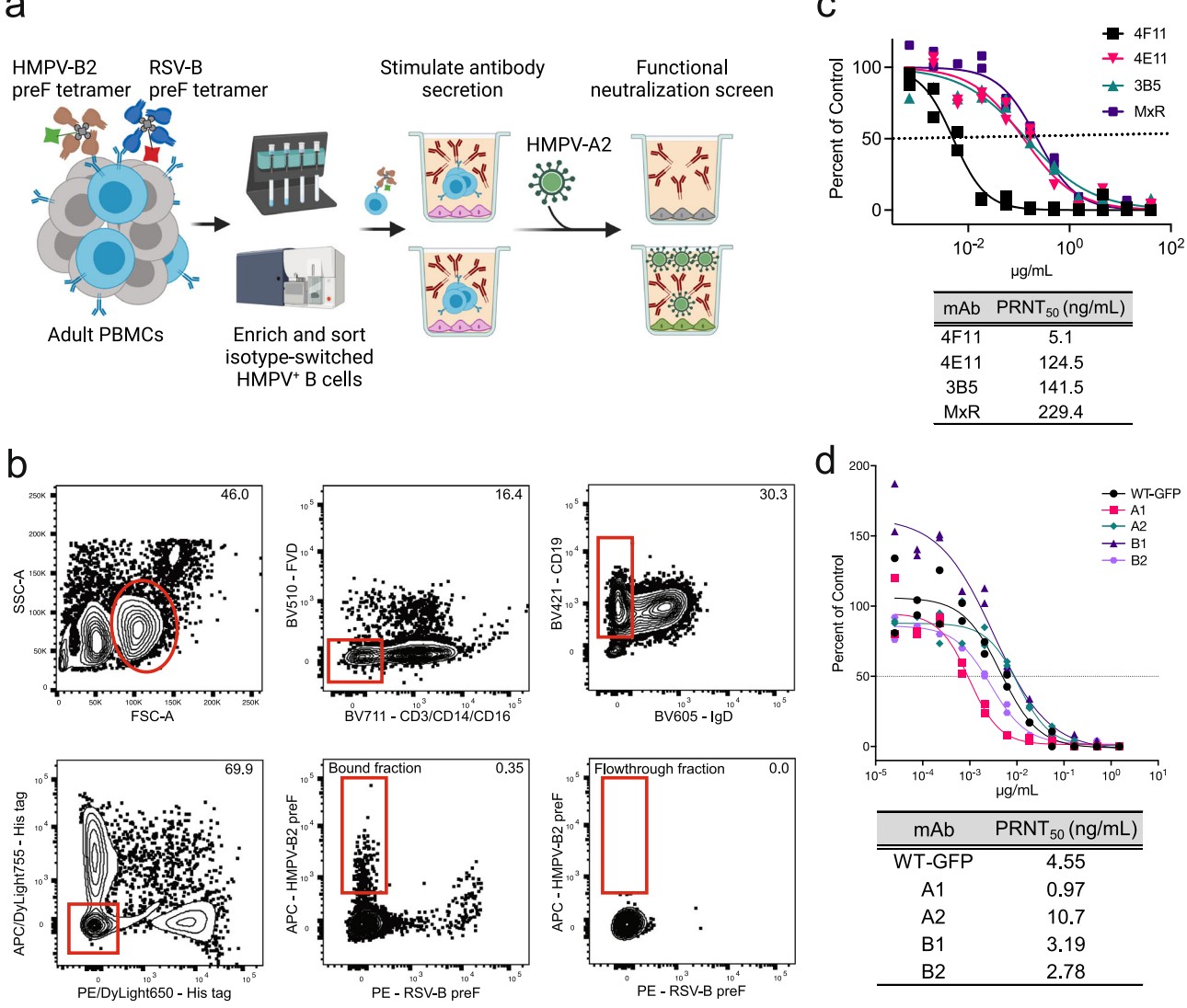

**Fig. 1 | Isolation of neutralizing human monoclonal antibodies targeting the F protein of HMPV. a** Bait-and-switch approach using an antigen from one HMPV strain to identify B cells that cross-neutralize another strain of the virus. HMPV B2-binding B cells from human peripheral blood and spleen were labeled with APC-conjugated tetramers of HMPV B2 preF and sorted. **b** Flow cytometry plot of HMPV B2 preF-binding B cells after gating for live, CD3⁻CD14⁻CD16⁻CD19⁺CD20⁺ (B cells), IgD⁻ (isotype-switched), and APC/Dylight755⁻His tag⁻ (to exclude cells binding to the His tag, APC, or streptavidin). PE-conjugated tetramers of RSV B preF and PE/DyLight650-conjugated control tetramers of the His tag were also included for comparison. The bound fraction contains cells magnetically enriched using APC- and PE-specific microbeads. The flowthrough fraction contains cells that did not bind the magnet and were thus depleted of antigen-specific cells. Numbers in plots are percentages of total cells in the gate. **c** Vero cells were infected with HMPV A2 in the presence of serial dilutions of the indicated mAbs. Data points are the average ± SD from two independent experiments. Neutralization potency was determined by a 50% plaque reduction neutralization test (PRNT$_{50}$), which is also indicated by the dotted line. **d** Vero cells were infected with HMPV subtypes A1, A2, B1, or B2 in the presence of serial dilutions of 4F11. WT-GFP indicates the same wild-type HMPV-A2 expressing a GFP reporter used in the neutralization assay shown in (**c**). The recombinant virus was generated based on the wild-type sequence of CAN97-83. The dotted midline represents the PRNT$_{50}$. Data points are the average ±SD of two replicates. **a** was created in BioRender. Boonyaratanakornkit, J. (2026) https://BioRender.com/rum3jsc.

monomer bound to two Fabs (one Fab each of MxR and 4F11) (Fig. S2). We obtained a 4.03 Å resolution cryoEM structure of 4F11 and MxR Fabs bound to the DS-CavEs2 P185A monomer (Figs. 2b and S3a–c, and Table S1). As predicted from the earlier binding competition experiments, the structure showed that 4F11 binds site Ø almost vertically at the apex of HMPV F in the prefusion conformation (Fig. 2b).

Since a structure of 4F11 bound to an intact trimer could not be obtained using DS-CavEs2, newer stabilized versions of HMPV preF were explored. Recombinant v3B is a prefusion-stabilized variant of HMPV F that includes amino acid substitutions V84C-A249C to generate interprotomer disulfide bonds, similar to another variant DS-CavEs2-IPDS[45,46]. Attempts to complex 4F11 Fab to the v3B trimer[45] with Pro185 reverted to alanine similarly resulted in trimer dissociation

upon the addition of 4F11 Fab, as observed with the GCN4-stabilized and DS-CavEs2 trimers. We next turned to the most recently published, stabilized version of the HMPV preF trimer, MPV-2c[47]. The MPV-2c trimer does not rely on stabilization with Pro185 and was co-transfected with 4F11 Fab due to low expression of the trimer. Fortunately, the MPV-2c trimer remained intact when co-transfected with 4F11 Fab. We obtained a 4.13 Å resolution cryoEM structure of this complex (Figs. 2c, S3d–f and S4, and Table S1). As expected, 4F11 binds site Ø at the apex of preF, confirming what was determined with the monomer structure (Fig. 2b), and only one Fab is bound per trimer (1:1 stoichiometry of 4F11 Fab:MPV-2c trimer). Indeed, when we model the trimer with two or three Fabs of 4F11 bound, the Fabs clash with each other. This suggests that there is structurally only enough room for

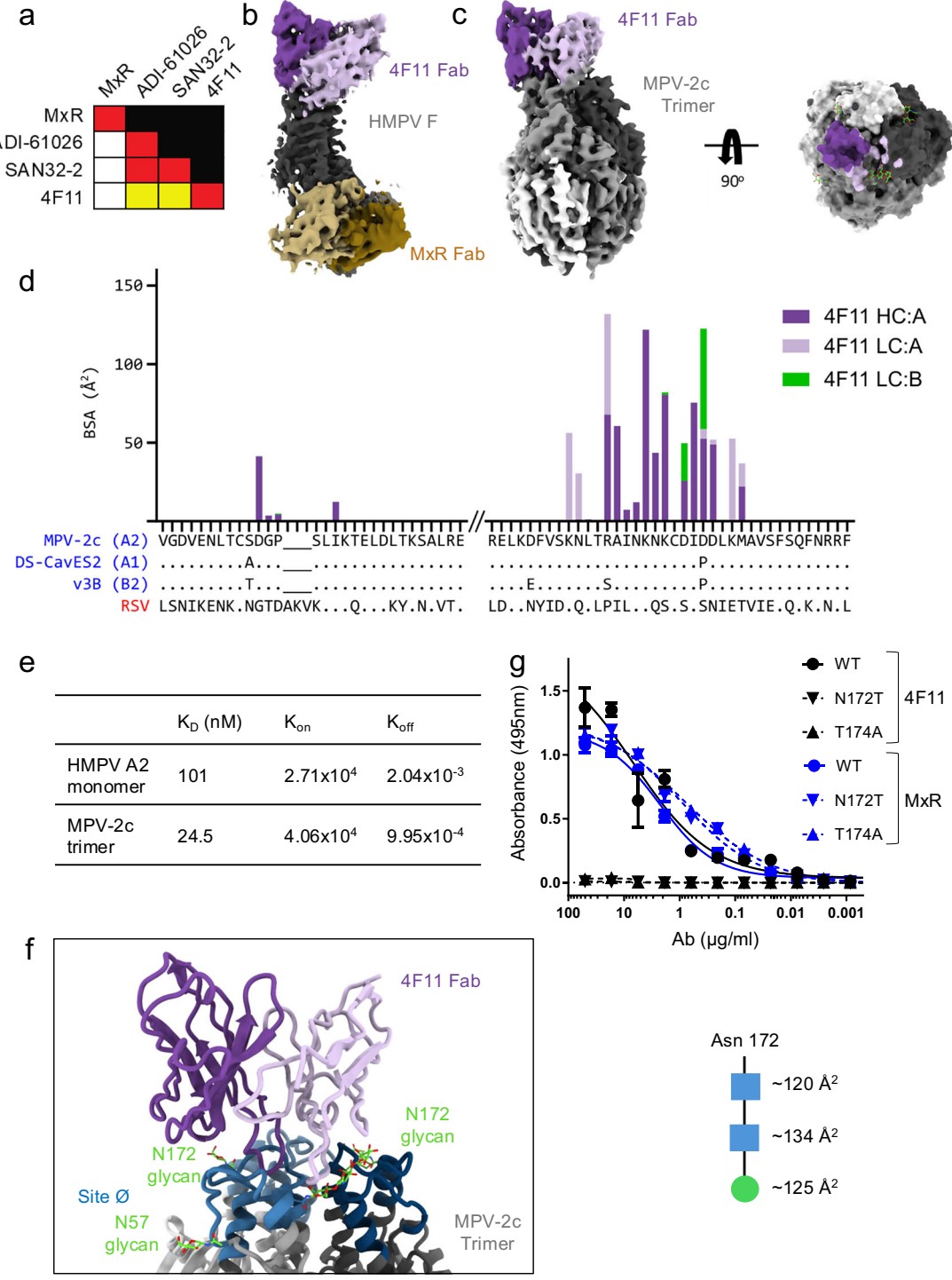

**Fig. 2 | Epitope mapping and structure of 4F11 binding HMPV F. a** Biolayer interferometry (BLI) measurements of the ability of the mAb listed on the left side of the chart to block binding of the mAb listed on the top. Competition is expressed as the percent drop in maximum signal compared to the maximum signal in the absence of competing mAb (red, 80–100%; yellow, 40–80%; white, 0–40%). **b** Cryo-EM map of 4F11 (heavy chain, purple; light chain, light purple) and MxR (heavy chain, brown; light chain, light brown) Fabs in complex with DS-CavES2 P185A monomer (gray). Only the VH/VL domains are shown. **c** Cryo-EM map of 4F11 (purple) Fab in complex with MPV-2c trimer (shades of gray) (left). Top view surface representation (right). The footprint of 4F11 VH is shown in dark purple and VL in light purple in the top view. **d** Plot showing buried surface area (BSA) contribution of protomer A and adjacent protomer B residues of the MPV-2c trimer when interacting with 4F11. Interactions with 4F11 VH are shown in dark purple (protomer A), and interactions with 4F11 VL are shown in light purple (protomer A) or green (protomer B). The sequences of MPV-2c, DS-CavES2, v3B, and RSV are aligned with the HMPV subtypes in parentheses. **e** Kinetics of 4F11 Fabs binding to HMPV A2 monomer and MPV-2c trimer by BLI. **f** Zoomed in view of the cartoon representation of MPV-2c F trimer and 4F11 Fab with site Ø shown in shades of blue and site Ø glycans in green (left). BSA of the N172 glycan with N-acetylglucosamine (NAG) represented as blue squares and mannose represented as a green circle (right). **g** Binding measured by ELISA of 4F11 and MxR IgG to HMPV trimer with and without the N172 glycan. Data points are the average ± SD from three independent experiments.

binding by one 4F11 Fab at a time to the trimer (Fig. S5). M8C10 is another mAb that causes trimer dissociation, but it binds an epitope that is buried deep within the trimerization interface[48]. M8C10 interferes with trimer formation likely by clashing with the non-binding protomers of the trimer (Fig. S6). In contrast, 4F11 likely facilitates trimer dissociation when more than one Fab tries to bind the trimer, because the Fabs clash with each other (Fig. S6).

4F11 binds site Ø with a total buried surface area (BSA) of ~895 Å$^2$, ~600 Å$^2$ of which are contributed by the heavy chain interaction with protomer A, ~220 Å$^2$ from the light chain interaction with protomer A, and ~75 Å$^2$ from the light chain interaction with protomer B (Figs. 2d and S7a, b). As expected, since 4F11 interacts with residues on two HMPV F protomers, 4F11 has higher affinity for the MPV-2c trimer than for the monomer (4-fold) (Figs. 2e and S8). When the F trimer transitions from the prefusion conformation to the postfusion conformation, site Ø undergoes large rearrangements that would cause 4F11 to clash with portions of the postfusion trimer (Fig. S9). Therefore, 4F11 preferentially binds a quaternary prefusion-specific epitope. Of the 4F11 residues that contact the trimer, only 3 out of 9 in the VH and 4 out of 17 in the VL are mutated from their germline genes, VH5-51*01 and VK2-28*01, respectively (Fig. S7b).

Site Ø is highly conserved across the four HMPV subtypes (Fig. S7a), which explains why 4F11 is able to neutralize all four subtypes (Fig. 1d). Since Asp185 is in the epitope of 4F11, the stabilizing Pro185 mutation in v3B and DS-CavEs2 likely disrupts the interactions between Asp185 and 4F11. Notably, 4F11 accommodates the glycan shield at site Ø that consists of the Asn57 and Asn172 glycans. While 4F11 avoids the Asn57 glycan by binding vertically towards the top of the trimer, its light chain interacts with the Asn172 glycan (glycan total BSA of ~379 Å$^2$) (Fig. 2f and S7b), which is necessary for binding since removal of the Asn172 glycan by mutagenesis or EndoH tretament abrogates 4F11 binding (Figs. 2g and S10a). Since contact by 4F11 with the glycan at Asn172 contributes only a fraction of the overall buried surface area, we would not expect 4F11 to bind non-specifically to other viral glycoproteins. Indeed, 4F11 does not bind to RSV preF (Fig. S10b), as expected, since our B cell sorting strategy already selected for B cells that could bind HMPV but not RSV preF (Fig. 1a, b). In contrast to 4F11, two other site Ø antibodies, SAN32-2[37] and ADI-61026[36] bind equatorially between the Asn57 and Asn172 glycans and penetrate the glycan shield without interacting with the glycans (Fig. S11). Thus, 4F11 is a quaternary-preferring, prefusion-specific, and glycan-dependent antibody with a unique binding mode, distinct from other site Ø antibodies.

## HMPV escape mutants

To select for viable escape mutations, we passaged HMPV A2 expressing a GFP reporter in the presence of serial dilutions of either 4F11 or no antibody as a negative control for lab-adapted mutations. We included MxR as a positive control, since substitutions at D280 in HMPV F have been previously reported as a mechanism of escape to site III mAbs[49]. By passage 3 in the presence of MxR, most cells were infected and expressing GFP even at 15 μg/mL of MxR, which is a concentration approximately 100-fold higher than its PRNT$_{50}$ (148 ng/mL)[42] (Fig. 3a). Deep sequencing revealed a mixed population consisting of viruses with mutations at S237P, D280N, D280G, and S483N (Fig. 3b). These mutations were introduced into recombinant HMPV F protein, and MxR lost binding to HMPV F with S237P, D280N, and D280G, but not S483N (Fig. 3c). Notably, the yield of recombinant F with the S237P mutation was substantially lower compared to wild-type F (36%) and F with the other escape mutations (Fig. 3b). We attempted to isolate viruses containing the individual escape mutations by plaque purification. Out of 22 plaques picked, 6 were cultured and sequenced successfully. We isolated a pure population of virus with D280G from 4 plaques. The D280N and S483N mutations co-occurred in the other 2 plaque-purified viruses, which is consistent

with their identical allele frequencies during passaging (Fig. 3b). We were unable to plaque purify virus with S237P, which had the lowest allele frequency during passaging, and which had a F protein that was difficult to produce recombinant.

In contrast to MxR, infection and GFP expression was negligible at concentrations of 4F11 at or above 470 ng/mL, even after 5 passages (Fig. 3a). At the next dilution of 235 ng/mL, less than 25% of cells were infected at passages 1 through 4, and the level of infection did not appreciably increase with each successive passage. Instead, GFP expression was lost by passage 5. Despite the low level of infection, viruses from the third passage were successfully expanded in the presence of 235 ng/mL of 4F11, and deep sequencing identified a K179E mutation that was present in 100% of the virus population (Fig. 3b). 4F11, but not MxR, lost binding to recombinant HMPV F protein containing the K179E mutation (Fig. 3c). In contrast, the mutations derived during HMPV passaging in the presence of MxR (S237P, D280N, D280G, and S483N) had no effect on 4F11 binding to recombinant HMPV F protein (Fig. 3c).

Following plaque purification, we successfully cultured and sequenced 5 out of 11 plaques. All 5 contained a pure population of virus with the K179E mutation, except for one, which also carried a new N466K mutation in the F protein, a mutation not detected in our initial metagenomic sequencing. The N466K mutation in isolation had no effect on 4F11 binding to recombinant HMPV F protein (Fig. 3d). The K179E and K179E + N466K viruses were then used to evaluate the in vitro neutralization potency of the 4F11 antibody and to assess replication kinetics relative to wild-type (WT) HMPV. The neutralizing potency of 4F11 was determined by calculating the PRNT$_{50}$ in Vero cells infected with WT, K179E, or K179E + N466K mutant viruses. 4F11 still neutralized HMPV carrying the K179E mutation but had a PRNT$_{50}$ of 192.3 ng/mL, compared to a PRNT$_{50}$ of 2.2 ng/mL to WT virus (Fig. 3e). The combined K179E + N466K mutant virus had an approximately 2-fold further increase in PRNT$_{50}$ to 430.5 ng/mL. To assess whether these mutations impact viral fitness, we compared viral growth kinetics in Vero cells. The K179E virus showed reduced growth kinetics compared to WT HMPV, with titers approximately 2 logs lower by day 7 post-infection (Fig. 3f). The virus containing both K179E and N466K together showed slightly improved growth kinetics relative to K179E alone, suggesting that while N466K does not mediate antibody escape, it may partially compensate for the fitness cost associated with K179E.

To investigate whether the K179E mutation is observed in circulating HMPV strains, we analyzed global sequence data using Nextstrain[50]. Among 782 publicly sequenced strains, 0 contained an E at position 179 (Fig. S12). At position 179, only 2 strains had a conservative replacement with a Q, 179 strains had an R, and the remaining 601 had a K. The structure of 4F11 bound to the HMPV F trimer revealed that Lys179 interacts with a negatively charged patch of 4F11 that includes residues in the heavy chain (Figs. 2d and 3g) and contributes to ~12% total BSA of HMPV F (Fig. 2d). The K179E mutation would introduce a negatively charged residue that would repel the negatively charged patch on 4F11 (Fig. 3g), providing a mechanism for this resistance mutation. Unlike K179E, which was not found in any publicly sequenced isolates of HMPV, replacement at position 179 with Q, R, or K residues, which are either neutral or positively charged, would not lead to repulsion.

Since 4F11 binding to HMPV F is glycan-dependent, mutation of the Asn172 glycan motif could represent another mechanism of escape. Although we failed to isolate any live virus with a mutation at glycosylation sites, we observed that 4F11 failed to bind recombinant MPV-2c trimer with an N172T mutation that removes the Asn172 glycan (Fig. 2g). The N172T substitution was chosen because we found a single HMPV F sequence with a threonine at position 172 in Nextstrain (Fig. S13). All other 998 sequences (99.9%) in Nextstrain had an asparagine.

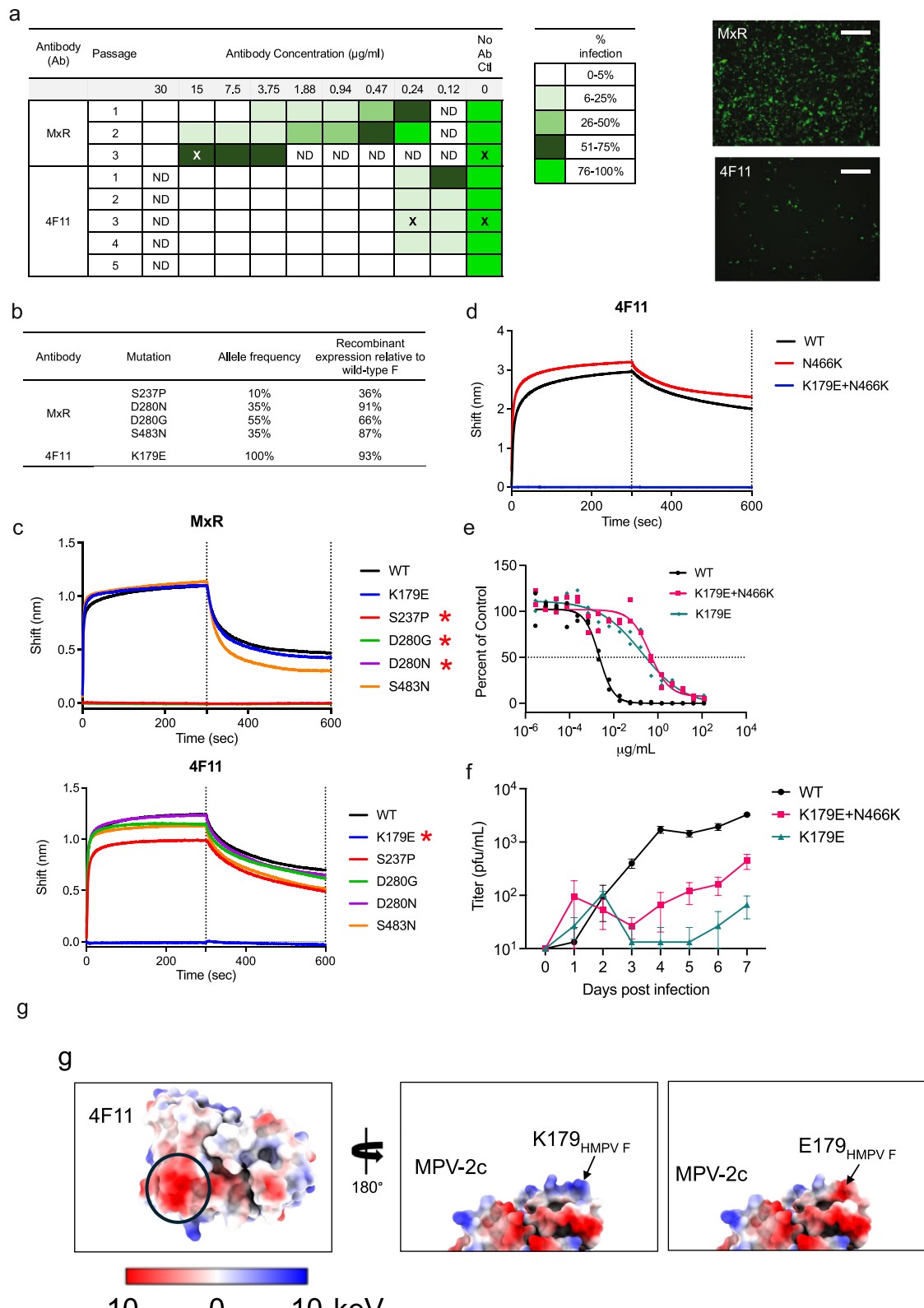

## In vivo protection

We next investigated whether the in vitro binding and neutralization data would translate into in vivo protection in an animal challenge model. The hamster model has been used extensively to evaluate vaccine and antibody candidates for HMPV[42,51–55]. Although some studies have observed higher titers of HMPV in the lungs of cotton rats compared to hamsters[56,57], this could be related to differences in the viral strain used, the size of the inoculum, and the timing of lung sampling. Since our primary objective was the development of a mAb potent enough to be used as therapy, we performed preclinical testing in the Golden Syrian hamster model in which 4F11 was administered post-infection. In treating respiratory viral infections with mAbs, the greatest clinical benefit is generally observed with early administration to prevent progression to severe disease[58,59]. Therefore, 4F11 was

**Fig. 3 | Escape mutation analysis. a** Passaging results after HMPV A2, expressing a GFP reporter, were mixed with serial dilutions of MxR, 4F11, or no antibody, as a control. An "X" indicates the sample from which virus was sequenced by metagenomics. Percent infection was determined by assessing GFP expression with fluorescence microscopy. ND, not done. Each passaging experiment was performed once. The entire well was visually scanned, and a representative fluorescence microscopy image is shown for the sample with MxR at 15 μg/mL at passage 3 (top) and for the sample with 4F11 at 0.24 μg/mL at passage 3 (bottom). The scale bar represents 500 μm. **b** Metagenomic sequencing of viruses. Mutations listed are those with at least 10% allele frequency and that were absent in the no antibody control samples. Recombinant HMPV F with the listed mutations was produced by transient transfection in 293 cells, and the yield of mutant F was compared to wild-type HMPV F. **c** Binding of MxR IgG (top) and 4F11 IgG (bottom) to recombinant HMPV preF protein was measured by biolayer interferometry. Measurements are normalized against an isotype control antibody. Asterisks indicate mutations that led to loss of binding by mAb. **d** Biolayer interferometry measurements of binding of 4F11 IgG to stabilized HMPV preF containing a N466K substitution in the presence or absence of a K179E substitution. **e** Vero cells were infected with HMPV-WT (wild-type), HMPV-K179E, or HMPV-K179E + N466K in the presence of serial dilutions of 4F11 antibody. The dotted midline represents the $PRNT_{50}$. Data points are the average ± SD of two replicates. **f** Viral growth kinetics of HMPV-WT, HMPV-K179E, and HMPV-K179E + N466K. Vero cells were infected at an MOI of 0.1. Cell culture supernatant was collected at each timepoint and quantified using a viral titer assay. Data points are the average ± SD of three replicates. **g** Electrostatics of 4F11 and HMPV F around amino acid position 179. The 4F11 paratope (left) is colored by electrostatic values. The binding site for K179 is circled in black. 180° rotation showing the MPV-2c trimer surface colored by electrostatic charge with K179 (middle) and E179 mutation (right).

administered intramuscularly to hamsters 24 h after intranasal inoculation with HMPV. Lungs and nasal turbinates were harvested on day 4 post-infection to determine the titer of HMPV in the lower and upper respiratory tracts, respectively (Fig. 4a). This dosing and route of administration are similar to those used in cotton rat models of HMPV prophylaxis, except that we administered mAb after infection instead of before infection[42,43]. At a dose of 2.5 mg/kg, 4F11 fully suppressed viral replication of HMPV in the lungs of 7 out of 10 hamsters and significantly reduced viral replication in nasal turbinates (Fig. 4b, c). The remaining 3 hamsters had viral titers at or near the limit of detection. At a reduced dose of 1.25 mg/kg, 4F11 fully suppressed viral replication of HMPV in 2 of 5 animals and still significantly reduced viral replication in the nasal turbinates. In contrast to 4F11, the cross-neutralizing mAb MxR failed to block viral replication in the lungs of any animals at a dose of 2.5 mg/kg (Fig. 4c). For comparison, RSV antibodies such as palivizumab and nirsevimab are clinically administered for prophylaxis in infants at doses higher than that used in the present study (15 mg/kg and approximately 10–20 mg/kg, respectively)[60–62]. Therefore, the dosage used in our in vivo experiments is lower than the dosages of other mAbs in clinical use, which is an important consideration from the standpoint of potential toxicity in humans.

## Discussion

There are no available treatment options for HMPV infection, and no mAbs are currently approved for clinical use against this virus. In the present study, we identified and characterized 4F11, a human mAb that exhibits high potency, neutralization of diverse strains of HMPV, and therapeutic efficacy in a hamster challenge model.

Others have observed binding to HMPV preF by 0.07–3.69% of class-switched B cells[39,43]. The expressed antibodies from 22 to 69% of these B cells had detectable neutralization of HMPV at concentrations up to 10 μg/mL. The majority of HMPV-neutralizing B cells (75%) were able to produce antibodies that neutralized both subtypes A and B[39]. In comparison, our study identified a relatively lower frequency (0.5%) of neutralizing B cells among isotype-switched HMPV preF-binding B cells, because our screening strategy was designed to functionally select for B cells that produce highly potent neutralizing antibodies upfront, without the need to sequence, clone, and produce mAbs from all HMPV F-binding B cells. This screening strategy leads to detectable IgG in the supernatant of approximately 40–65% of wells with individually sorted B cells[42,63]. Of the wells with detectable IgG, approximately 35–100% have antibodies in culture supernatant that bind to the same antigen used as the B cell probe. The median IgG level is approximately 4 ng/mL, although some B cells can produce over 475 ng/mL[63]. Therefore, this screening strategy favors the selection of neutralizing antibodies with an $IC_{50}$ in the ng/mL range, like 4F11, and tends to miss weaker neutralizing antibodies. Furthermore, the recombinant GCN4-HMPV F used as the B cell probe is known to contain a mixture of prefusion and postfusion conformations of HMPV

F[40]. Therefore, HMPV postF-binding B cells were also likely individually sorted into wells and contributed to the relatively lower frequency of neutralizing B cells observed in our screen.

Our structural analysis revealed that 4F11 employs a unique antiviral mechanism that sets it apart from previously characterized mAbs against HMPV and RSV. Specifically, 4F11 targets an epitope located at the apex of the prefusion conformation of the F protein (site Ø) with a 1:1 stoichiometry of 4F11 Fab: preF trimer. This stoichiometry at site Ø has not been previously described for mAbs targeting HMPV but has been observed with tmAbs targeting the preF protein of RSV[64] and HPIV3[65]. Unlike previously described site Ø antibodies against HMPV preF, such as SAN32-2[39] and ADI-61026[43], which bind with a 3:1 stoichiometry and penetrate the glycan shield between Asn57 and Asn172, 4F11 binds vertically and directly interacts with the Asn172 glycan. This unique glycan-dependent mode of recognition separates 4F11 from other site Ø antibodies. 4F11 is also able to bind both monomeric and trimeric F, but has a higher affinity to trimeric F due to contacts with the adjacent protomer by the light chain. This makes 4F11 quaternary preferring, which has similarly been observed with HIV mAbs, such as PG9[66]. Since 4F11 binds to site Ø and is prefusion-specific, 4F11 likely neutralizes HMPV by preventing the conformational switch of F from a prefusion to postfusion state, a step essential for viral entry.

We also noticed dissociation of the WT trimer with the GCN4 trimerization domain, the DS-CavEs2 trimer, and the v3B trimer when they were complexed with 4F11. The HMPV prefusion F-specific mAb M8C10 also disrupts trimer formation, similar to 4F11[48]. However, unlike 4F11, which approaches vertically to bind a quaternary epitope at the apex, the M8C10 epitope is buried deep within the core of the trimer, near antigenic sites II and III, that may only be accessible when the trimer is widely open during the process of "breathing". Compared to 4F11 ($IC_{50}$ range 1.0–10.7 ng/mL), M8C10 has relatively weaker neutralization potency ($IC_{50}$ range 371–661 ng/mL). Neutralizing antibodies targeting the trimer interface of type 1 fusion proteins, like M8C10 for HMPV and several anti-HA antibodies for influenza, can neutralize virus by disrupting the trimer, thereby preventing subsequent fusion and viral entry[67–69]. Therefore, it is possible that trimer dissociation by 4F11 may be another mechanism that contributes to in vitro neutralization and in vivo efficacy.

There are currently several versions of prefusion-stabilized, trimeric HMPV F. Many of these, such as DS-CavEs2 and v3B, have a Pro185 mutation that is located within the 4F11 epitope at the apex of F. This stabilizing mutation prevents 4F11 binding, likely due to increased rigidity of the F trimer and the removal of a residue that interacts with 4F11. We successfully identified 4F11 in our antibody discovery campaign because we had serendipitously used HMPV F without the Pro185 stabilizing mutation. GCN4-stabilized HMPV F was first described in 2012, and newer, more stable constructs of HMPV F have since been reported, including DS-CavEs2 in 2022, v3B in 2023, and MPV-2c in 2024[40,45,47,70]. We used GCN4-stabilized HMPV F as the probe in our B cell sorting experiments, because this was the only recombinant

 

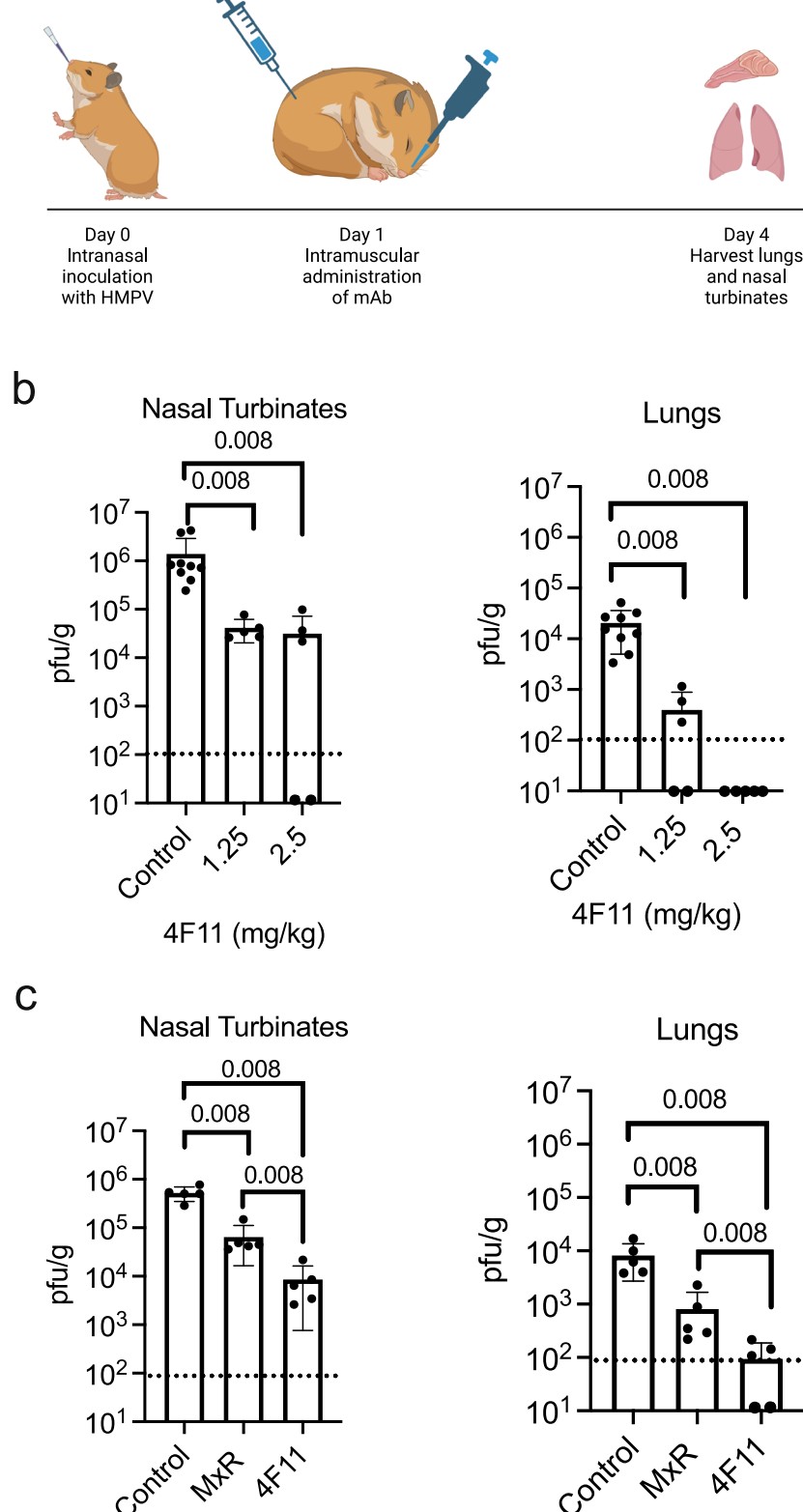

**Fig. 4 | In vivo efficacy of 4F11 as therapy for HMPV. a** Schematic of experiments in which hamsters were injected intramuscularly with mAb, 24 h after intranasal challenge with $10^5$ pfu of HMPV A2. **b** Dose–response of 4F11 on HMPV replication ($n = 5$ experimental animals/dose, $n = 9$ control animals/dose). **c** Comparison of HMPV replication in hamsters treated with 2.5 mg/kg of 4F11 or MxR mAb ($n = 5$ animals per group). Viral titers were measured by plaque assay in nasal homogenates (left) and lungs (right) from individual hamsters at 4 days post-infection. Dashed lines indicate the limit of detection. Bars represent the mean, error bars represent standard deviations, and *p* values are calculated by two-sided Mann–Whitney test. Control hamsters were injected with 1× DPBS. **a** was created in BioRender. Boonyaratanakornkit, J. (2026) https://BioRender.com/zbugyzr.

version of HMPV F reported at the time to adopt a prefusion conformation. Structural analysis of GCN4-HMPV F has revealed that a mixture of prefusion and postfusion conformations is present, indicating that GCN4-HMPV F is relatively less stable compared to newer versions of recombinant HMPV F[40]. However, if newer constructs with the Pro185 mutation had been used to sort B cells, then 4F11-like antibodies would have been missed. The presence of this mutation in vaccine candidates would fail to induce the maturation of B cells that target the 4F11 epitope, since immunogens with Pro185 would not be able to bind 4F11-like B cell receptors. The glycan shield of HMPV F contributes to the rarity of neutralizing mAbs targeting the apex[39,43,71], and the A185P mutation is protected by this glycan shield. Since no other neutralizing mAb to HMPV has previously been described to bind vertically at the apex, the 4F11 epitope is subdominant. Although we have shown that 4F11-like antibodies can be very potent, they may not make a major contribution to vaccine immunogenicity due to their rarity. Further studies comparing vaccine candidates with and without the A185P mutation will be needed to assess potential differences in vaccine immunogenicity. Nevertheless, the introduction of stabilizing mutations within neutralizing epitopes, such as Pro185, could lead to vaccines that miss an opportunity to elicit a subset of potent neutralizing mAbs. Interestingly, we found that relatively few residues of 4F11 that contact the trimer are mutated from their germline sequences. Therefore, 4F11-like B cell receptors would not require extensive somatic hypermutation to achieve high neutralization potency. Moreover, it is possible that further affinity maturation could boost the neutralization potency of 4F11.

Monoclonal antibodies have shown considerable promise as both prophylactic and therapeutic agents against respiratory viruses, with multiple mAbs approved or authorized for clinical use against pathogens such as RSV and SARS-CoV-2[20,72]. However, a major roadblock for antibody-based therapies is the emergence of resistance, which can limit long-term efficacy. Herein, we characterized potential resistance mutations to 4F11 by passaging HMPV in vitro under constant antibody pressure. We identified a single escape mutation, K179E, that introduces a negatively charged residue in F at a key 4F11 interaction site that is also negatively charged. Functional studies revealed that this mutation significantly impaired 4F11's binding and neutralizing ability but also reduced viral growth, indicating a fitness cost that comes with this mutation. A secondary mutation, N466K, appears to have an epistatic effect by partially compensating for the fitness cost associated with K179E. These 2 amino acids, however, are on opposite ends of the trimer and do not interact in either the prefusion or postfusion conformation. Further biophysical and higher resolution structural characterization of 4F11 in complex with F, F with K179E alone, and F with K179E in combination with N466K are needed to elucidate the mechanisms of neutralization escape, growth attenuation, and partial compensation. While we identified an escape mutation in vitro, it is important to note that K179E has not been observed in circulating clinical strains globally. In our in vitro passaging experiments, the K179E escape mutation was maintained in the setting of continuous 4F11 mAb pressure, raising the possibility that the K179E mutation would revert and disappear without selective pressure. Future studies will investigate the stability of the K179E mutation in the absence of continuous mAb pressure. Although the prefusion stabilizing Pro185 mutation in DS-CavEs2 and v3B also abrogated binding by 4F11, Pro185 is unlikely to represent a viable escape mutation since it prevents the transition of F from the prefusion to the postfusion conformation, a process that is essential for infection. Mutation of the glycosylation motif at Asn172 represents another potential route of escape. However, only 1 out of 999 (0.1%) sequences on Nextstrain have an N172T substitution. This rarity in circulating HMPV strains likely reflects a substantial fitness cost of the mutation. Indeed, site-directed mutagenesis and reverse genetics were used to recover

recombinant HMPV lacking glycosylation at position 172, and this virus was profoundly attenuated in vitro and in vivo[73].

In vivo, we focused our efforts on evaluating the potential of using 4F11 as a therapeutic agent. Our hamster challenge model demonstrated that a 2.5 mg/kg dose of 4F11 administered 1 day after infection fully suppressed viral replication in the lungs of most hamsters and significantly reduced replication in the nasal turbinates. Even at a lower dose of 1.25 mg/kg, viral loads were substantially decreased, with complete suppression of replication in the lungs observed in 2 of 5 hamsters. We chose to evaluate 4F11 in a treatment model due to its high potency and specificity for HMPV. Although the therapeutic window for treatment of HMPV infections with mAbs is currently undefined, the COVID-19 pandemic provided several examples of mAbs that met endpoints as post-exposure prophylaxis or early treatment of symptomatic disease in phase III clinical trials. For instance, sotrovimab, when administered within 5 days of symptom onset, reduced the risk of hospitalization or death among high-risk patients with mild to moderate COVID-19 (NCT04545060)[74]. The cocktail of antibodies casirivimab and imdevimab (REGN-COV2), when administered within 7 days of symptom onset, also reduced hospitalization or death and led to more rapid symptom resolution and reduction of SARS-CoV-2 viral load among patients with risk factors for severe disease (NCT04425629)[75]. This cocktail also demonstrated efficacy as post-exposure prophylaxis of symptomatic disease when administered to participants within 4 days after a household contact tested positive for SARS-CoV-2 (NCT04452318)[76]. These examples provide proof-of-concept that administration of mAbs within a few days of exposure or symptom onset can prevent respiratory viral disease. Future clinical trials will be needed to define the therapeutic window for HMPV. Notably, similar to 4F11, sotrovimab and REGN-COV2 were formulated as IgG1 with preserved Fc effector function. The relative contribution of Fc effector function compared to neutralization for protection against disease in the setting of treatment may be dose-dependent[77,78]. For instance, in a hamster challenge model of SARS-CoV-2 infection, a clinically relevant dose of a Fc receptor binding null variant of a potent neutralizing antibody had equal potency to wild-type or Fc receptor binding enhanced variants, indicating that Fc-independent neutralization was the primary mode of protection. However, at lower doses, Fc effector functions contributed to decreased viral replication and decreased weight loss. Future studies in hamsters will be needed to define the relative contribution of Fc effector functions to protection at varying doses of 4F11. While prophylaxis remains critical for immunocompromised individuals who may not respond well to vaccines, effective treatment options are also important for improving outcomes in the setting of an established infection. Given its high potency, 4F11 would also likely demonstrate effectiveness as prophylaxis. In contrast to 4F11, mAbs like MxR[42], MPE8[79], and 25P13[80] exhibit broader reactivity to RSV and HMPV, making them more suitable for prophylactic use when the specific viral threat is unknown, and cross-protection is advantageous. Alternatively, the 4F11 mAb could be mixed with an RSV-specific mAb to achieve dual protection against RSV and HMPV.

Taken together, our findings establish 4F11 as a promising candidate for clinical development against HMPV infection. Its high potency, neutralizing activity against diverse strains, unique glycan-dependent binding mechanism, low resistance susceptibility, and significant in vivo therapeutic efficacy support its further advancement. 4F11 may provide a much-needed therapeutic option for vulnerable populations, including infants, the elderly, and immunocompromised individuals, who are most at risk for severe HMPV disease.

## Methods

### Study design

This study complies with all relevant ethical regulations and was reviewed and approved by the Fred Hutchinson Cancer Center

Institutional Review Board (IRB). Peripheral blood was obtained by venipuncture from healthy, HIV-seronegative adult volunteers enrolled in the Seattle Area Control study after informed consent (Protocol #5567). PBMCs were isolated from whole blood using Accuspin System Histopaque-1077 (Sigma-Aldrich, cat#10771). Studies involving human spleens were deemed non-human subjects research since tissue was de-identified. Spleen samples received an exempt determination by the Fred Hutchinson Institutional Review Board, as defined by the Common Rule, from the Office for Human Research Protections. Tissue was de-identified and originated from deceased donors in which the spleen would have otherwise been discarded during procurement of other organs (e.g., liver) for donation. Tissue fragments were passed through a basket screen, centrifuged at $300 \times g$ for 7 min, incubated with ACK lysis buffer (Thermo Fisher, cat#A1049201) for 3.5 min, resuspended in RPMI (Gibco, cat#11875093), and passed through a stacked 500 µm and 70 µm cell strainer. Cells were resuspended in 10% dimethylsulfoxide in heat-inactivated fetal calf serum (Gibco, cat#16000044) and cryopreserved in liquid nitrogen before use.

## Cell lines

293F cells (Thermo Fisher, cat#R79007) were cultured in Freestyle 293 media (Thermo Fisher, cat#12338026). Vero cells (ATCC CCL-81) and LLC-MK2 cells (ATCC CCL-7.1) were cultured in DMEM (Gibco, cat#12430054) supplemented with 10% fetal bovine serum and 100 U/mL penicillin plus 100 µg/mL streptomycin (Gibco, cat#15140122).

## Viruses

The recombinant virus HMPV-GFP has been previously described[81] and was modified from HMPV CAN97-83 (GenBank accession number AY297749) to express enhanced GFP. HMPV was cultured on LLC-MK2 cells with DMEM (Gibco, cat#12430054) supplemented with 100 U/mL penicillin plus 100 µg/mL streptomycin (Gibco, cat#15140122) and 1.2% of 0.05% Trypsin (Gibco, cat#25300054). Virus was purified by centrifugation in a discontinuous 30%/60% sucrose gradient with 0.05 M HEPES and 0.1 M $MgSO_4$ (Sigma-Aldrich, cat#H4034 and 230391, respectively) at $120,000 \times g$ for 90 min at 4 °C. Virus titers were determined by infecting Vero cell monolayers in 24-well plates with serial 10-fold dilutions of virus, overlaying with DMEM containing 0.8% methylcellulose (Sigma-Aldrich, cat#M0387) and 2.4% of 0.05% Trypsin (Gibco, cat#25300054). Fluorescent plaques were counted using a Typhoon scanner (GE Life Sciences) at 5 days post-infection.

Clinical strains of HMPV were obtained from ZeptoMetrix, including A1 (strain IA10-2003, lot#331204, cat#0810161CF), A2 (strain IA27-2004, lot#331149, cat#0810164CF), B1 (strain Peru3-2003, lot#328937, cat#0810158CF), and B2 (strain IA18-2003, lot#329066, cat#0810162CF). Plaques were counted by fixing wells with 10% formalin for 1 h, washing three times with 1× PBS with 0.02% Tween 20 (PBST), and then incubating with 1× DPBS containing 10% goat serum (Southern Biotech, cat#0060-01) for 1 h at room temperature. Wells were washed three times with 1× PBST and then incubated with the 4F11 antibody for 1 h. After 3 additional washes with 1× PBST, wells were incubated for 1 h with Alexa Fluor 647-conjugated goat anti-human IgG at a dilution of 1:2000 (Invitrogen, cat#A-21445) and scanned using a Typhoon scanner (GE Life Sciences).

## Expression and purification of antigens

Expression plasmids for His-tagged RSV preF and HMPV preF are previously described[40,45,47,82]. 293F cells were transfected at a density of $10^6$ cells/mL in Freestyle 293 media using 1 mg/mL PEI Max (Polysciences, cat#24765). Transfected cells were cultured for 7 days with gentle shaking at 37 °C. Supernatant was collected by centrifuging cultures at $2500 \times g$ for 30 min followed by filtration through a 0.2 µm filter. The clarified supernatant was incubated with Ni Sepharose beads overnight at 4 °C, followed by washing with wash buffer containing 50 mM Tris, 300 mM NaCl, and 8 mM imidazole. His-tagged protein was eluted with an elution buffer containing 25 mM Tris, 150 mM NaCl, and 500 mM imidazole. The purified protein was run over a 10/300 Superose 6 size exclusion column (GE Life Sciences, cat#17−5172−01). Fractions containing the trimeric F proteins were pooled and concentrated by centrifugation in a 50 kDa Amicon ultrafiltration unit (Millipore, cat#UFC805024). The concentrated sample was stored in 50% glycerol at −20 °C.

For structural characterization, MPV-2c trimer was co-transfected with 4F11 His-Fab and Furin in a 3:1:1:0.6 (MPV-2c: 4F11 His-Fab HC: 4F11 His-Fab LC: Furin) ratio in HEK293E cells at a density of $10^6$ cells/mL using 2 mL of 1 mg/mL PEI Max per liter of cells transfected. Transfected cells were cultured for 6 days at 37 °C with shaking. Supernatant was collected by centrifuging and then filtering the cell cultures using a 0.2 µm filter. The supernatant was incubated with Ni-NTA resin (Takara Bio) for 1 h at room temperature with shaking, followed by washing with wash buffer containing 50 mM Tris, 15 mM imidazole, and 0.3 M NaCl at pH 8.0 and eluted with an elution buffer containing 50 mM Tris, 150 mM imidazole, and 0.3 M NaCl at pH 8.0. The complex was then further purified with size exclusion chromatography (SEC) over a Superdex200 16/600 on an AKTApure system (GE Biosciences) equilibrated in 1× Tris-buffered saline (TBS). The final complex was concentrated to 2 mg/mL with a 10 kDa cutoff Amicon ultrafiltration unit (Millipore, cat#UFC8010).

## Tetramerization of antigens

Purified F antigens were biotinylated using an EZ-link Sulfo-NHS-LC-Biotinylation kit (Thermo Fisher, cat#A39257) with a 1:1.3 molar ratio of biotin to F. Unconjugated biotin was removed by centrifugation using a 50 kDa Amicon Ultra size exclusion column (Millipore, cat# UFC805024). To determine the average number of biotin molecules bound to each molecule of F, streptavidin-PE (ProZyme, cat#PJRS25) was titrated into a fixed amount of biotinylated F at increasing concentrations and incubated at room temperature for 30 min. Samples were run on an SDS-PAGE gel (Invitrogen, cat#NW04127-BOX), transferred to nitrocellulose, and incubated with streptavidin−Alexa Fluor 680 (Thermo Fisher, cat#S32358) at a dilution of 1:10,000 to determine the point at which there was excess biotin available for the streptavidin−Alexa Fluor 680 reagent to bind. Biotinylated F was mixed with streptavidin-allophycocyanin (APC) or streptavidin-PE at the ratio determined above to fully saturate streptavidin and incubated for 30 min at room temperature. Unconjugated F was removed by centrifugation using a 300 K Nanosep centrifugal device (Pall Corporation, cat#OD300C33). APC/DyLight755 and PE/DyLight650 tetramers were created by mixing F with streptavidin-APC pre-conjugated with DyLight755 (Thermo Fisher, cat#62279) or streptavidin-PE pre-conjugated with DyLight650 (Thermo Fisher, cat#62266), respectively, following the manufacturer's instructions. On average, APC/DyLight755 and PE/DyLight650 contained 4−8 DyLight molecules per APC and PE. The concentration of each tetramer was calculated by measuring the absorbance of APC (650 nm, extinction coefficient = 0.6 $\mu M^{-1} cm^{-1}$) or PE (566 nm, extinction coefficient = 2.0 $\mu M^{-1} cm^{-1}$).

## Tetramer enrichment

$1–2 \times 10^8$ frozen PBMCs or spleen cells were thawed into DMEM with 10% fetal calf serum and 100 U/mL penicillin plus 100 µg/mL streptomycin. Cells were centrifuged and resuspended in 50 µL of ice-cold fluorescence-activated cell sorting (FACS) buffer composed of phosphate-buffered saline (PBS) and 1% fetal calf serum (Fisher Scientific, cat#SH3007103). His tag APC/DyLight755 and PE/Dylight650-conjugated tetramers were added at a final concentration of 25 nM in the presence of 2% rat and mouse serum (Thermo Fisher) and incubated at room temperature for 10 min. PreF APC and PE tetramers were then added at a final concentration of 5 nM and incubated on ice for 25 min, followed by a 10 mL wash with ice-cold FACS buffer.

Next, 50 μL each of anti-APC-conjugated (Miltenyi Biotec, cat#130-090-855) and anti-PE-conjugated (Miltenyi Biotec, cat#130-048-801) microbeads were added and incubated on ice for 30 min, after which 3 mL of FACS buffer was added and the mixture was passed over a magnetized LS column (Miltenyi Biotec, cat#130-042-401). The column was washed once with 5 mL ice-cold FACS buffer and then removed from the magnetic field, and 5 mL ice-cold FACS buffer was pushed through the unmagnetized column twice using a plunger to elute the bound cell fraction.

## Flow cytometry

Cells were incubated in 50 μL of FACS buffer containing a cocktail of antibodies for 30 min on ice prior to washing and analysis on a FACS Aria (BD). Antibodies included anti-CD19 BUV395 (SJ25C1, BD, cat#563551, 1:20 dilution), anti-CD3 BV711 (UCHT1, BD, cat#563725, 1:50 dilution), anti-CD14 BV711 (M0P-9, BD, cat#563372, 1:50 dilution), anti-CD16 BV711 (3G8, BD, cat#563127, 1:50 dilution), anti-IgD BV605 (IA6-2, BD, cat#563313, 1:50 dilution), and a fixable viability dye (Tonbo Biosciences, cat#13−0870−T500, 1:250 dilution). B cells were individually sorted into flat-bottom 96-well plates containing feeder cells that had been seeded at a density of 28,600 cells/well 1 day prior in 100 μL of IMDM media (Gibco, cat#31980030) containing 10% fetal calf serum, 100 U/mL penicillin plus 100 μg/mL streptomycin, and 2.5 μg/mL amphotericin. B cells sorted onto feeder cells were cultured at 37 °C for 13 days.

## B cell receptor sequencing

For individual B cells sorted onto feeder cells, supernatant was removed after 13 days of culture, plates were immediately frozen on dry ice, stored at −80 °C, thawed, and RNA was extracted using the RNeasy Micro Kit (Qiagen, cat#74034)[42,83,84]. The entire eluate from the RNA extraction was used instead of water in the RT reaction. Following RT, 2 μL of cDNA was added to 19 μL PCR reaction mix so that the final reaction contained 0.2 μL (0.5 U) HotStarTaq Polymerase (Qiagen, cat#203607), 0.075 μL of 50 μM 3′ reverse primers, 0.115 μL of 50 μM 5′ forward primers, 0.24 μL of 25 mM dNTPs, 1.9 μL of 10× buffer (Qiagen), and 16.5 μL of water. The PCR program was 50 cycles of 94 °C for 30 s, 57 °C for 30 s, and 72 °C for 55 s, followed by 72 °C for 10 min for heavy and kappa light chains. The PCR program was 50 cycles of 94 °C for 30 s, 60 °C for 30 s, and 72 °C for 55 s, followed by 72 °C for 10 min for lambda light chains. After the first round of PCR, 2 μL of the PCR product was added to 19 μL of the second-round PCR reaction so that the final reaction contained 0.2 μL (0.5 U) HotStarTaq Polymerase, 0.075 μL of 50 μM 3′ reverse primers, 0.075 μL of 50 μM 5′ forward primers, 0.24 μL of 25 mM dNTPs, 1.9 μL 10× buffer, and 16.5 μL of water. PCR programs were the same as the first round of PCR. 5 μL from the PCR reactions was mixed with 2 μL of ExoSAP-IT (Thermo Fisher, cat#78201) and incubated at 37 °C for 15 min followed by 80 °C for 15 min to hydrolyze excess primers and nucleotides. Hydrolyzed second-round PCR products were sequenced by Genewiz with the respective reverse primer used in the second-round PCR, and sequences were analyzed using IMGT/V-Quest to identify V, D, and J gene segments. Paired heavy chain VDJ and light chain VJ sequences were cloned into pTT3-derived expression vectors containing the human IgG1, IgK, or IgL constant regions using In-Fusion cloning (Clontech, cat#638911)[85].

## Monoclonal antibody and Fab production

Secretory IgG and His-Fabs were produced by co-transfecting HEK293E cells at a density of $10^6$ cells/mL with the paired heavy and light chain expression plasmids at a ratio of 1:1 in Freestyle 293 media using 1 mg/mL PEI Max. Transfected cells were cultured for 6 days with gentle shaking at 37 °C. Supernatant was collected by centrifuging cultures at $2500 \times g$ for 15 min followed by filtration through a 0.2 μm filter. Clarified supernatants of secretory IgG were then incubated with

Protein A agarose (Thermo Scientific, cat#20333) followed by washing with IgG binding buffer (Thermo Scientific, cat#21007). Antibodies were eluted with IgG Elution Buffer (Thermo Scientific, cat#21004) into a neutralization buffer containing 1 M Tris-base pH 9.0. Purified antibody was concentrated and buffer exchanged into PBS using an Amicon ultrafiltration unit with a 50 kDa molecular weight cutoff for IgG. His-Fabs were purified with Ni-NTA affinity resin (Takara Bio, cat#635661) followed by SEC on a Superdex200 16/600 column (Cytiva). Fractions containing His-Fabs were then pooled and concentrated using an Amicon ultrafiltration unit with a 10 kDa cutoff.

## Bio-layer interferometry

Bio-layer interferometry (BLI) assays were performed on the Octet.Red instrument (ForteBio) at room temperature with shaking at 500 rpm. For binding analyses, anti-human IgG Fc capture sensors (ForteBio, cat#18−5060) were loaded in kinetics buffer (PBS with 0.01% bovine serum albumin, 0.02% Tween 20, and 0.005% $NaN_3$, pH 7.4) containing 40 μg/mL mAb for 150 s. After loading, the baseline signal was recorded for 60 s in kinetics buffer. The sensors were then immersed in kinetics buffer containing 30 μM purified HMPV preF for a 300 s association step, followed by immersion in kinetics buffer for a dissociation phase of 300 s. For competitive binding assays, anti-penta His capture sensors were loaded in kinetics buffer containing 1 μM His-tagged GCN4-stabilized HMPV preF for 300 s. After loading, the baseline signal was recorded for 30 s in kinetics buffer. The sensors were then immersed for 300 s in kinetics buffer containing 40 μg/mL of the first antibody, followed by immersion for another 300 s in kinetics buffer containing 40 μg/mL of the second antibody. Percent competition was determined by dividing the maximum increase in signal of the second antibody in the presence of the first antibody by the maximum signal of the second antibody alone.

## ELISA

Nunc MaxiSorp 96-well plates (Thermo Fisher, cat#442404) were coated with 100 ng of HMPV preF for 90 min at 4 °C. Wells were washed three times with 1× DPBS and then incubated with 1× DPBS containing 1% bovine serum albumin (Sigma-Aldrich, cat#A2153) for 1 h at room temperature. Antigen-coated plates were incubated with serial dilutions of mAb for 90 min at 4 °C. Wells were washed three times with 1× DPBS followed by incubation for 1 h with horseradish peroxidase-conjugated goat anti-human total Ig at a dilution of 1:6000 (Invitrogen, cat#31412). Wells were then washed four times with 1× DPBS followed by a 5–15 min incubation with TMB substrate (SeraCare, cat#5120−0053). Absorbance was measured at 405 nm using a Softmax Pro plate reader (Molecular Devices).

## Neutralization assays

For neutralization screening of culture supernatants, Vero cells were seeded in 96-well flat-bottom plates and cultured for 48 h. After 13 days of culture, 40 μL of B cell culture supernatant was mixed with 25 μL of sucrose-purified GFP-HMPV A2 diluted to 2000 plaque-forming units (pfu)/mL for 1 h at 37 °C. Vero cells were then incubated with 50 μL of the supernatant/virus mixture for 1 h at 37 °C to allow viral adsorption. Next, each well was overlaid with 100 μL DMEM containing 0.8% methylcellulose and 1.2% of 0.05% trypsin. Fluorescent plaques were counted at 5 days post-infection using a Typhoon imager.

Neutralizing titers of monoclonal antibodies were determined by a 50% plaque reduction neutralization test ($PRNT_{50}$). Vero cells were seeded in 24-well plates and cultured for 48 h. Monoclonal antibodies were serially diluted 1:4 in 120 μL DMEM and mixed with 120 μL of sucrose-purified HMPV diluted to 2000 pfu/mL for 1 h at 37 °C. Vero cells were incubated with 100 μL of the antibody/virus mixture for 1 h at 37 °C to allow viral adsorption. Each well was then overlaid with 500 μL DMEM containing 0.8% methylcellulose. Fluorescent plaques

were counted at 5 days post-infection using a Typhoon imager. $PRNT_{50}$ titers were calculated by linear regression analysis.

## CryoEM complex and grid preparation

The complex of DS-CavEs2 P185A monomer bound to 4F11 and MxR Fabs was made by incubating DS-CavEs2 P185A trimer with a 1:3.2 molar ratio of trimer: MxR Fab for 5 min at room temperature with shaking, then adding a 1:1.2 molar ratio of trimer: 4F11 Fab and incubating for 10 min with shaking at room temperature. The complex was then purified using a Superdex200 10/300 column (Cytiva) to remove excess Fab, and fractions containing DS-CavEs2 P185A monomer bound to MxR and 4F11 Fabs were collected. The complex of DS-CavEs2 P185A monomer bound to 4F11 and MxR Fabs was frozen on UltrAuFoil 1.2/1.3 Au 300 mesh grids (Ted Pella, cat#688-300-AU-50) using a Vitrobot Mk. IV (Thermo Fisher) at 4 °C and 100% humidity. Dodecyl-β-D-maltoside (DDM) was added to the sample before freezing at a final concentration of 0.1× CMC, and the grids were glow-discharged using a plasma cleaner. The final grids used for collection were frozen with 3 μL of complex at 3.5 mg/mL, a 5 s wait time, a 10 s blot time, and a blot force of 1. The grid of the complex of MPV-2c trimer and 4F11 Fab was prepared similarly, except the final grids used for collection were frozen with 3 μL of complex at 2 mg/mL, a 30 s wait time, a 6 s blot time, and a blot force of 2.

## Data collection, processing, and model refinement

Data for the complex of DS-CavEs2 P185A monomer bound to 4F11 and MxR Fabs were collected on a Glacios (Thermo Fisher) operating at 200 kV with a Gatan K3 DED. Data were collected using Serial EM at 92,000× magnification (1.122 Å/pix) and 60 e/Å$^2$ exposure dose to get 1048 micrographs. Data was processed using CryoSPARC v4.3.1 after being motion corrected with WARP[86]. After importing the motion corrected micrographs, patch CTF estimation was performed. Micrographs were then curated to include <7 Å CTF fit and to remove micrographs from thick ice. Blob picker was used to select 1,332,453 particles that were then extracted at 128px box size before undergoing multiple rounds of 2D classification. The 101,166 particles in the final set of good 2D classes were re-extracted at 256px box size. A single ab-initio class was generated. This map was further refined using homogeneous refinement and non-uniform refinement with C1 symmetry. A custom mask was then used to crop out the CH1 domain of the 4F11 Fab to produce a map of 4.03 Å resolution. Model refinement was performed in the Phenix software suite, Coot, and the ISOLDE plugin of ChimeraX. The cryoEM map was deposited to the EMDB. No model was built to fit this cryoEM map. All figures were produced in ChimeraX.

Data for the complex of MPV-2c trimer and 4F11 Fab was collected on a Glacios (Thermo Fisher Scientific) operating at 200 kV with a Gatan K3 DED. Data were collected using Serial EM at 92,000× magnification (1.122 Å/pix) and 60 e/Å$^2$ exposure dose to get 3327 micrographs. Data was processed using CryoSPARC v4.3.1 after being motion corrected with WARP[86]. After importing the motion corrected micrographs, patch CTF estimation was performed. Micrographs were then curated to include <7 Å CTF fit and to remove micrographs from thick ice. Blob picker was used to select 3,524,733 particles that were then extracted at 180px box size before undergoing multiple rounds of 2D classification. The 276,088 particles in the final set of good 2D classes were re-extracted at 360px box size. A single ab-initio class was generated. This map was further refined using homogeneous refinement and non-uniform refinement with C1 symmetry. A custom mask was then used to crop out the CH1 domain of the 4F11 Fab to produce a map of 4.13 Å resolution. Model refinement was performed in the Phenix software suite, Coot, and the ISOLDE plugin of ChimeraX. The cryoEM map and model were deposited to the PDB and EMDB. All figures were produced in ChimeraX.

## Escape mutation studies

Vero cells were seeded in 24-well flat-bottom plates and cultured for 48 h. Antibody was serially diluted 1:2 starting at 30 μg/mL in 500 μL of DMEM (Gibco, cat#11965092) supplemented with 100 U/mL penicillin, 100 μg/mL streptomycin (Gibco, cat#15140122), and 1.2% of 0.05% trypsin (Gibco, cat#25300054). HMPV was diluted to $1 \times 10^6$ pfu/mL, and 500 μL was mixed with each antibody dilution and incubated at 37 °C for 1 h. Virus/antibody mixtures were added to Vero cells, and viral replication was monitored by GFP expression. After 7 days, supernatants were collected and incubated with fresh antibody at 37 °C for 1 h, then used to inoculate new Vero cells in 24-well plates. After 3 passages for MxR and 5 passages for 4F11, wells containing the highest antibody concentration with visible viral replication were selected for expansion. From each selected well, 200 μL of supernatant was incubated with 2 mL of DMEM containing the same antibody concentration present in that well. After 1 h at 37 °C, the mixture was used to infect Vero cells in T-25 flasks. After 5–7 days in culture, virus was harvested for sequencing.

## Plaque purification

Vero cells were plated in 24-well plates and inoculated with serial 1:10 dilutions of virus, overlaid with methylcellulose containing 1.2% trypsin (Gibco, cat#25300054), and cultured for 5–7 days. Individual plaques were identified with fluorescent microscopy and picked with a P200 pipette tip. The aspirate containing the plaque was transferred onto Vero cells in 96-well plates containing 0.234 μg/mL of 4F11. After 7 days, supernatant was transferred to Vero cells in a T25 flask containing 0.234 μg/mL of 4F11. Purified virus was re-sequenced by Sanger sequencing.

## Metagenomic next-generation sequencing and data interpretation

RNA was extracted from a 500 μL volume of cell culture supernatant using the Quick-RNA Viral Kit (Zymo Research, cat#R1034). Extracted RNA was treated with Turbo DNA-free kit (Invitrogen, cat#AM1907) to remove genomic DNA. First-strand cDNA synthesis was performed using random hexamers (Invitrogen, cat#N8080127) and SuperScript IV enzyme (Invitrogen, cat#18090200), and second-strand synthesis was performed using Sequenase Version 2.0 (Applied Biosystems, cat#70775Z1000UN). The cDNA was purified using 1.0× AMPure XP beads (Beckman Coulter, cat#A63881) and quantified on a Qubit 3.0 (LifeTech). Purified cDNA was sent to Azenta Life Sciences, where it underwent tagmentation and library preparation, followed by metagenomic Next-Generation Sequencing as 2×150-bp runs on a NovaSeq X Plus (Illumina).

A Snakemake (v8.14) pipeline (available at https://github.com/jbloomlab/Virus-Antibody-Passaging/tree/main) was used to process sequencing reads and identify viral mutations that emerged during passaging[87]. The pipeline first preprocesses the unaligned reads with fastp (v0.23) using default settings to automatically identify and trim adapter sequences and filter low-quality bases[88]. The processed reads from the sequenced viral stock were then used to assemble a reference genome by aligning to the HMPV CAN97-83 sequence (AY297749) with BWA mem (v0.7), followed by generating a consensus sequence from the most frequent base at each position using iVar (v1.4.3)[89,90]. The remaining processed sequencing data— comprising two replicates each of 4F11-passaged virus and MxR-passaged virus, plus an untreated control—were aligned to the stock consensus using BWA mem. Following alignment, variants were identified using iVar with a minimum base quality threshold of 20. To remove potential false positives, heuristic filters were applied such that final reported variants must exceed 3% frequency, occur at sites covered by more than 200 reads, and be assigned a statistically significant p-value of <0.05 by iVar.

## Animals and viral challenge

All procedures were reviewed and approved by the Fred Hutch Institutional Animal Care and Use Committee (PROTO202300013) and conducted in accordance with institutional and National Institutes of Health guidelines. Golden Syrian hamsters (*Mesocricetus auratus*) were infected intranasally with 100 μL of $10^5$ pfu HMPV. Sample sizes ($n = 38$) with at least 5 animals per group were consistent with previously published experiments testing the efficacy of RSV mAbs in the cotton rat model[60,91–93]. Only male animals (4–8 weeks of age) were assessed in this work, because no association between sex and clinical outcomes has been observed in adults with RSV or HMPV[94,95]. Monoclonal antibody was administered intramuscularly at 2.5 or 1.25 mg/kg in 50 μL PBS, 24 h after infection. Nasal turbinates and lungs were removed for viral titration by plaque assay 4 days post-infection and clarified by centrifugation in DMEM. Confluent Vero cell monolayers were inoculated in duplicate with diluted homogenates in 24-well plates. After incubating for 1 h at 37 °C, wells were overlaid with 0.8% methylcellulose (made with 1.2% of 0.05% trypsin). After 5 days, plaques were counted using the Typhoon imager to determine titers as pfu per gram of tissue.

## Statistical analysis

Statistical analysis was performed using GraphPad Prism 10. Pairwise statistical comparisons were performed using Mann–Whitney two-tailed testing. $p < 0.05$ was considered statistically significant. Data points from individual samples are displayed.

## Reporting summary

Further information on research design is available in the Nature Portfolio Reporting Summary linked to this article.

## Data availability

Sequencing and structural data that support the findings of this study have been deposited in the Protein Data Bank (PDB) and Electron Microscopy Data Bank (EMDB) and are accessible through accession numbers 9ORE, EMD-70773, and EMD-70774. Viral sequences have been deposited in the Sequence Read Archive (SRA) and are accessible through BioProject PRJNA1413375, study SRP666688, accession numbers SRR36992303, SRR36996228, SRR36996229, and SRR36996230 [https://trace.ncbi.nlm.nih.gov/Traces/?view=study&acc=SRP666688]. Source data are provided with this paper.

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

## Acknowledgements

The authors thank Julie McElrath for PBMCs from the Seattle Area Control cohort; Andrew McGuire for providing CD40L/IL2/IL21-expressing 3T3 cells; Ramasamy Bakthavatsalam and LifeCenter Northwest for providing de-identified spleen remnants; Ursula Buchholz, Shirin Munir, and Peter Collins for providing the GFP-expressing HMPV; Peter Kwong and Ou Li for v3B expression constructs; Theodore Jardetzky and Xiaolin Wen for GCN4-stabilized expression constructs of HMPV preF; Steve Voght for proof-reading the manuscript; Andrea Langenhuizen, Elizabeth McCarthy, and Laura Yates for administrative support; Fred Hutch Flow Cytometry and Cellular Imaging Shared Resources for assistance with instruments; Fred Hutch Comparative Medicine Shared Resources for assistance with housing hamsters; and the members of the Boonyaratanakornkit Lab for helpful discussions. Electron microscopy data were generated using the Fred Hutch Cancer Center Electron Microscopy shared resource (EMSR). The EMSR is supported in part by the Cancer Center Support Grant P30 CA015704-40. This research was also supported by the Cellular Imaging Shared Resource RRID:SCR_022609 of the Fred Hutch/University of Washington/Seattle Children's Cancer Consortium [P30 CA015704]. Experimental schematics were created with BioRender.com. This study was supported by the Vaccine and Infectious Disease Division Faculty Initiative (J.B.) and Evergreen Beyond Pilot Award (J.B.) from the Fred Hutchinson Cancer Center, a sponsored research agreement with IgM Biosciences (J.B.), a New Investigator Award from the American Society for Transplantation and Cellular Therapy (J.B.), the Amy Strelzer Manasevit Award from the National Marrow Donor Program (J.B.), and NIH R01-AI171186 (J.B.). The content is solely the responsibility of the authors and does not necessarily represent the official views of the National Institutes of Health.

## Author contributions

E.H. designed and conducted the experiments, analyzed the data, and wrote the manuscript. S.P., R.I., L.K., M.G., and S.H. conducted experiments and analyzed data. M.M. and M.P. coordinated and performed the structural analysis and wrote the manuscript. C.-N.C. and T.-M.F. conceived the study, designed experiments, and analyzed the data. J.B. conceived the study, designed experiments, conducted experiments, analyzed the data, and wrote the manuscript. A.G., E.S., and W.H. designed the viral sequencing pipeline. All authors edited the manuscript.

## Competing interests

J.B. is an inventor of patent applications filed by Fred Hutchinson Cancer Center directed to the 4F11 and MxR antibodies. S.H., C.-N.C., and T.-M.F. were employees of IgM Biosciences. The remaining authors declare no competing interests.
