## [Peer Review file · Nature Communications]

DEVELOPMENT OF A POTENT MONOCLONAL ANTIBODY FOR TREATMENT OF HUMAN METAPNEUMOVIRUS INFECTIONS

Corresponding Author: Dr Jim Boonyaratankornkit

Version 0:

Reviewer comments:

Reviewer #1

(Remarks to the Author)

This manuscript is exceptionally well written and presents significant findings on a novel human monoclonal antibody, 4F11, targeting hMPV. The authors describe 4F11 as binding to site 0 at the apex of the hMPV F protein, with structural characterization revealing a unique angle of Fab engagement that spans across protomers and strongly interacts with the N-linked glycan at position 172. These features clearly distinguish 4F11 from previously characterized site 0 antibodies, SAN32-2 and ADI-61026. Furthermore, the authors raise an important concern regarding the A185P mutation, which may affect vaccine immunogenicity. This insight has meaningful implications for rational vaccine design and underscores the importance of accounting for such mutations in future immunogen development.

Overall, the manuscript makes a valuable contribution to the field and provides novel insights that could inform both therapeutic antibody development and vaccine strategies against hMPV.

There are three major concerns related to the structural and biophysical characterization of Fab 4F11 in complex with hMPV F.

- Model building could be largely improved given all the refinement tools available today (Coot, Phenix and Isolde). In supplemental table 1, the clashscore ideally should be below 5 and Ramachandran favored regions should be easily refined to be greater than 95%. In addition, rotamer outliers can be lower than 1%.
- We recommend adding a supplemental figure showing the map-to-model fit at the interface. This would help assess sidechain placement and contacts and provide strong support for the interpretation shown in Figure 2f.
- In supplemental figure 4, authors may include both experimental binding curves and fitted curves. What fitting model was used? Without all this information, kinetics data shown in figure 2e might not be accurate.

In addition, there are a few minor comments and would appreciate the authors' thoughts on these points.

- The authors raised a concern about A185P mutation potentially affecting vaccine immunogenicity. Do you have any supporting evidence showing that 4F11 type of antibodies is immunodominant? Or has this type of antibodies turned out to be rare?
- In the protection study shown in Figure 4, the authors used DPBS as a control and tested two doses of 4F11 in a therapeutic setting, showing reduced virus loads in both the upper and lower respiratory tracts. While the results are encouraging, including a well-known monoclonal antibody as a comparator (e.g., MxR, MPE8, etc.) would make the findings stronger. This would help show how 4F11 performs relative to existing antibodies.
- Based on the MS, 4F11 appears to be designed for therapeutic use, given its high potency, broad cross-neutralizing activity, and resistance to escape mutations. However, it still faces a fundamental challenge common to therapeutic settings: once infection is established, much of the disease severity is driven by the host immune response rather than ongoing viral replication. We would appreciate hearing the authors' perspective on this issue. Specifically, how do you envision overcoming this limitation to achieve meaningful therapeutic benefit in humans?

(Remarks on code availability)

Reviewer #2

(Remarks to the Author)

Harris et al. discovered a potent neutralizing antibody 4F11 against human metapneumovirus (hMPV). The approach is to use single B cell sorting to isolate naturally occurring human monoclonal antibodies from human donors who have been previously infected naturally. The bait and switch strategy of binding to hMPV B2 preF antigen and screening for neutralization against hMPV A2 appears robust in this study for discovery of cross-neutralizing Abs, although the conservation of F antigen between the two subtypes are high and it is expected that anti-F antibody should cover both subtypes. The authors characterized 4F11 using several biochemical and structural analysis, including cryoEM. A novel binding mode for the antibody to interact with hMPV F antigen site 0 was identified and the binding is also glycan dependent. Interesting, 4F11 binds to hMPV preF trimer at a 1:1 ratio rather than 3:1, and appears to disrupt the trimer formation in some in vitro conditions. The authors further characterized the potency of the antibody using in vitro neutralization assays and resistance selection, as well as in vivo hamster challenge model, demonstrating its potential application for hMPV therapy.

The manuscript is overall well written. Antibodies against RSV F has been extensively studied, and recently a few publications also described profiling of antibody response against hMPV. However, 4F11 appears to be a novel class of site 0 antibody against hMPV which has not been identified previously, making this manuscript a valuable addition for the field to further understand immune response against hMPV.

Major Comments:

The authors should consider further discuss potential (unique) mechanism of this antibody to neutralize hMPV. It is possible that the antibody disassociates hMPV prefusion F and/or prevent the conformational change of prefusion to postfusion conformation. The author should discussion another hMPV antibody described in the literature M8C10 (Structural characterization of M8C10, a neutralizing antibody targeting a highly conserved prefusion-specific epitope on the metapneumovirus fusion trimerization interface - PubMed) as it also binds the trimer interface although the epitope appears to be different than 4F11. There are a few trimer disassociating antibodies described in the literature against other viruses as well (such as flu) and can be included in discussion as a general theme.

The authors suggest that the primary goal is to develop an antibody for therapeutic indication against hMPV. In that case, the role of Fc should be discussed and evaluated in in vitro (such as Fc mediated assays) and/or in vivo studies. The authors performed in vivo experiments to demonstrate the protection efficacy of these antibodies as a therapy. While the data did suggest that these antibodies could offer protection, the lack of benchmark antibodies in the in vivo study prevents comparison of these antibodies to others previously described in the field.

Specific Comments:

P1 line 21 and 31: should not use the word "broad" or "broadly" here and throughout. hMPV F is highly conserved between the two subtypes, it is expected anti-F antibodies cross-react to both subtypes and different strains. In fact, this is true for most previously described hMPV F antibodies.

P1 line 40-43: should indicate disease burden is also high in older adults. Also, should clarify the approved RSV vaccines are for older adults, while there are two approved RSV monoclonal antibodies indicated for infants.

P2 line 40: "subtypes" instead of "strains".

P2 line 45: I am not convinced the construct (GCN4) used in Wen et al. is stabilized in prefusion conformation. Authors should discuss rationale for this antigen choice in the B cell sorting experiment.

P3 line 10-11: surprised to see only 6 out of 1140 clones neutralize hMPV A2. How many of them neutralize B strain? How many of them bind preF from A strain? Authors should discuss whether the majority of the B cells binds preF but couldn't neutralize hMPV.

P5 line 38: is there a structural explanation as to how N466K could compensate K179E?

P6 line 17 and 21: I believe the correct figure cited should be Fig 4, not Fig 5.

P6 line 24-27: the dose comparison with the cited previous studies is apple to orange, as they were for prophylaxis. Especially the last sentence on the clinical dose of palivizumab and nirsevimab is not relevant here.

P6 line 36: please clarify the the "1:1" stoichiometry is Ab to preF trimer.

P21 line 16: "WT indicates the same WT hMPV A2 expressing a GFP reporter" – it is confusing that "WT" actually refers to a GFP reporter virus, suggest to rephrase.

(Remarks on code availability)

I can't access the website provided: <https://github.com/jbloomb/lab/Virus-Antibody-Passaging/tree/main>

Reviewer #3

(Remarks to the Author)

(Remarks on code availability)

Reviewer #4

(Remarks to the Author)

The use of 1:1 stoichiometry vs 3:1 stoichiometry terminology used throughout the paper is not correct. hMPV F is a homotrimer, so the majority of antibodies discovered previously are actually 1:1 (1 Fab per 1 F monomer), and it is 3:1 in this case.

Is it possible that two or three Fabs could bind? Provide a size exclusion curve comparing this complex to that of one where other previously discovered Fabs binds in a 1:1 ratio.

Given the very high sequence conservation between hMPV A and B genotypes, the bait and switch methodology appears to be overstated. Based on the hundreds of antibodies previously isolated for hMPV F, it appears most neutralize both genotypes anyways.

Is the original isotype of the sequenced B cells that produced the monoclonal antibodies known?

Page 4 line 1: The hMPV F protein is known to dissociate into monomers easily, it is not appropriate to say that 4F11 "caused" the trimer to dissociate. However, please provide the data for this in the supplemental section.

The resolution of the cryoEM structure is low, which limits the significance of the work. Perhaps trying to complex with MPE8 and 4F11 would help improve the resolution, since MPE8 binds across protomers. Alternatively, complexing 4F11 with DsCavEs2-IPDS could work since this construct has interprotomer disulfide bonds.

A map to model image of the whole complex and interaction site should be provided given the low resolution.

Does 4F11 bind other viral glycoproteins similar to antibodies like 2G12, given its interaction with glycan residues? At a minimum, provide binding data with pre-fusion RSV F or some other negative control antigen.

Is the glycan important for binding? N172T is used, but why was threonine chosen and not aspartate?

Plaque purification of viral clones is mentioned in the results, but the methods don't mention picking individual plaques, how was this done?

There are no cells in the last gated population in Figure 1 b.

In supplemental 4, provide the raw data as well as the fit curves on top in the figures.

(Remarks on code availability)

Version 1:

Reviewer comments:

Reviewer #1

(Remarks to the Author)

Thank you for addressing our three major concerns from the previous review (reviewer #1). However, after careful consideration of the revised manuscript and the authors' responses, we still have reservations about all three major concerns that require further attention.

For Major concerns 1 and 2

We appreciated authors made efforts to show the model-to-fit interface in FigS4 and figured out the heavy and light chains should flip. However, cryo-EM map seems not meet the high quality for the high-impact factor journal and dataset seems suffers from orientation bias. I will encourage the authors collect another cryo-EM dataset to achieve higher resolution, which enables accurate model building. If glycan N172 plays a key role in contacting with 4F11 Fab, then authors should also show the map and model at this binding interface instead of only showing glycan alone on F.

For major concern 3

We appreciated authors added fit and experimental curves for BLI. However, KD value of 4F11 Fab bound to monomer in the table seems incorrect if KD is truly derived from kon and koff.

(Remarks on code availability)

Reviewer #2

(Remarks to the Author)

The revised manuscript has addressed my comments.

(Remarks on code availability)

Reviewer #3

(Remarks to the Author)

(Remarks on code availability)

Reviewer #4

(Remarks to the Author)

The authors addressed all of my concerns.

(Remarks on code availability)

We appreciate the generally positive comments from the the Reviewers. We would like to thank the reviewers for their detailed feedback and are grateful for the opportunity to respond. The manuscript has been revised, and the point-by-point response is found below. Reviewer comments are presented in a numbered outline (1, 2, 3), with our responses provided indented below each comment and labeled alphabetically (a, b, c). Page and line numbers refer to the manuscript document in which changes are tracked. Of note, during the initial review period, we realized that the heavy and light chain structures of our 4F11 antibody were flipped. We rebuilt our model of 4F11 with the correct heavy and light chains fitted for the cryo-EM map. We regret not catching this mistake earlier. Importantly, this correction reinforces and does not undermine the overall message of our paper. These changes are also tracked and a new PDB validation report has been provided.

REVIEWER COMMENTS

Reviewer #1 (Remarks to the Author):

This manuscript is exceptionally well written and presents significant findings on a novel human monoclonal antibody, 4F11, targeting hMPV. The authors describe 4F11 as binding to site 0 at the apex of the hMPV F protein, with structural characterization revealing a unique angle of Fab engagement that spans across protomers and strongly interacts with the N-linked glycan at position 172. These features clearly distinguish 4F11 from previously characterized site 0 antibodies, SAN32-2 and ADI-61026. Furthermore, the authors raise an important concern regarding the A185P mutation, which may affect vaccine immunogenicity. This insight has meaningful implications for rational vaccine design and underscores the importance of accounting for such mutations in future immunogen development. Overall, the manuscript makes a valuable contribution to the field and provides novel insights that could inform both therapeutic antibody development and vaccine strategies against hMPV. There are three major concerns related to the structural and biophysical characterization of Fab 4F11 in complex with hMPV F.

1. Model building could be largely improved given all the refinement tools available today (Coot, Phenix and Isolde). In supplemental table 1, the clashscore ideally should be below 5 and Ramachandran favored regions should be easily refined to be greater than 95%. In addition, rotamer outliers can be lower than 1%.
 - a. We are grateful to the reviewer for their highly positive feedback. We fixed the model to have a clash score less than 5, Ramachandran favored regions greater than 95%, and rotamer outliers below 1% (**Table S1**).
2. We recommend adding a supplemental figure showing the map-to-model fit at the interface. This would help assess sidechain placement and contacts and provide strong support for the interpretation shown in Figure 2f.

- a. We have added a new **Fig. S4** showing the map-to-model fit at the interface.
3. In supplemental figure 4, authors may include both experimental binding curves and fitted curves. What fitting model was used? Without all this information, kinetics data shown in figure 2e might not be accurate.
 - a. We have added the fitted curves to **Fig. S8**. The use of a 1:1 fitting model is now mentioned in the figure legend.
4. In addition, there are a few minor comments and would appreciate the authors' thoughts on these points. The authors raised a concern about A185P mutation potentially affecting vaccine immunogenicity. Do you have any supporting evidence showing that 4F11 type of antibodies is immunodominant? Or has this type of antibodies turned out to be rare?
 - a. The reviewer brings up an important point about immunodominance and the potential impact of the A185P-stabilizing mutation on vaccine immunogenicity. We have included a more in-depth discussion regarding this point on page 8, lines 33-41: "The glycan shield of HMPV F contributes to the rarity of neutralizing monoclonal antibodies targeting the apex (Rush, et. al., *Cell Rep*, 2022; Rappazzo et. al., *Immunity*, 2022; Guo, et. al., *Lancet Microbe*, 2023), and the A185P mutation is protected by this glycan shield. Since no other neutralizing monoclonal antibody to HMPV has previously been described to bind vertically at the apex, the 4F11 epitope is subdominant. Although we have shown that 4F11-like antibodies can be extremely potent, they may not make a major contribution to vaccine immunogenicity due to their rarity. Further studies comparing vaccine candidates with and without the A185P mutation will be needed to assess potential differences in vaccine immunogenicity. Nevertheless, the introduction of stabilizing mutations within neutralizing epitopes, such as Pro185, could lead to vaccines that miss an opportunity to elicit a subset of potent neutralizing mAbs."
5. In the protection study shown in Figure 4, the authors used DPBS as a control and tested two doses of 4F11 in a therapeutic setting, showing reduced virus loads in both the upper and lower respiratory tracts. While the results are encouraging, including a well-known monoclonal antibody as a comparator (e.g., MxR, MPE8, etc.) would make the findings stronger. This would help show how 4F11 performs relative to existing antibodies.
 - a. We thank the reviewer for this feedback. As suggested, we have now included a new **Fig. 4c** comparing the in vivo therapeutic efficacy of 4F11 with the cross-neutralizing monoclonal antibody MxR. We have discussed these new results on page 7, lines 8-9: "In contrast to 4F11, the cross-

neutralizing mAb MxR failed to block viral replication in the lungs of any animals at a dose of 2.5 mg/kg (**Fig. 4c**).”

6. Based on the MS, 4F11 appears to be designed for therapeutic use, given its high potency, broad cross-neutralizing activity, and resistance to escape mutations. However, it still faces a fundamental challenge common to therapeutic settings: once infection is established, much of the disease severity is driven by the host immune response rather than ongoing viral replication. We would appreciate hearing the authors’ perspective on this issue. Specifically, how do you envision overcoming this limitation to achieve meaningful therapeutic benefit in humans?
 - a. The reviewer raises a good point. We have included a more in-depth discussion regarding this point on page 9, lines 34-46 and page 10, line 1: “Although the therapeutic window for treatment of HMPV infections with monoclonal antibodies is currently undefined, the COVID-19 pandemic provided several examples of monoclonal antibodies that met endpoints as post-exposure prophylaxis or early treatment of symptomatic disease in phase III clinical trials. For instance, sotrovimab, when administered within 5 days of symptom onset, reduced the risk of hospitalization or death among high-risk patients with mild to moderate COVID-19 (NCT04545060) (Gupta, et. al., *N Engl J Med*, 2021). The cocktail of antibodies casirivimab and imdevimab, when administered within 7 days of symptom onset, also reduced hospitalization or death and led to more rapid symptom resolution and reduction of SARS-CoV-2 viral load among patients with risk factors for severe disease (NCT04425629) (Weinreich, et. al., *N Engl J Med*, 2021). This cocktail also demonstrated efficacy as post-exposure prophylaxis of symptomatic disease when administered to participants within 4 days after a household contact tested positive for SARS-CoV-2 (NCT04452318) (O’Brien, et. al., *N Engl J Med*, 2021). These examples provide proof-of-concept that administration of monoclonal antibodies within a few days of exposure or symptom onset can prevent respiratory viral disease. Future clinical trials will be needed to define the therapeutic window for HMPV.”

Reviewer #2 (Remarks to the Author):

Harris et al. discovered a potent neutralizing antibody 4F11 against human metapneumovirus (hMPV). The approach is to use single B cell sorting to isolate naturally occurring human monoclonal antibodies from human donors who have been previously infected naturally. The bait and switch strategy of binding to hMPV B2 preF antigen and screening for neutralization against hMPV A2 appears robust in this study for discovery of cross-neutralizing Abs, although the conservation of F antigen between the two subtypes are high and it is expected that anti-F antibody should cover both subtypes. The authors characterized 4F11 using several biochemical and structural analysis, including cryoEM. A novel binding mode for the antibody to interact with hMPV F antigen site 0 was identified and the binding is also glycan dependent. Interesting, 4F11 binds to hMPV preF trimer at a 1:1 ratio rather than 3:1, and appears to disrupt the trimer formation in some in vitro conditions. The authors further characterized the potency of the antibody using in vitro neutralization assays and resistance selection, as well as in vivo hamster challenge model, demonstrating its potential application for hMPV therapy. The manuscript is overall well written. Antibodies against RSV F has been extensively studied, and recently a few publications also described profiling of antibody response against hMPV. However, 4F11 appears to be a novel class of site 0 antibody against hMPV which has not been identified previously, making this manuscript a valuable addition for the field to further understand immune response against hMPV.

1. Major Comments: The authors should consider further discuss potential (unique) mechanism of this antibody to neutralize hMPV. It is possible that the antibody disassociates hMPV prefusion F and/or prevent the conformational change of prefusion to postfusion conformation. The author should discussion another hMPV antibody described in the literature M8C10 (Structural characterization of M8C10, a neutralizing antibody targeting a highly conserved prefusion-specific epitope on the metapneumovirus fusion trimerization interface - PubMed) as it also binds the trimer interface although the epitope appears to be different than 4F11. There are a few trimer disassociating antibodies described in the literature against other viruses as well (such as flu) and can be included in discussion as a general theme.
 - a. We are grateful to reviewer #2 for their positive comments and helpful feedback. A potential mechanism of neutralization by 4F11 is stabilization of the prefusion conformation to prevent the transition to the postfusion conformation. We have added a new **Fig. S9** demonstrating this. We describe this figure on page 4, lines 40-43: "When the F trimer transitions from the prefusion conformation to the postfusion conformation, site Ø undergoes large rearrangements that would cause 4F11 to clash with portions of the postfusion trimer (**Fig. S9**).” We have also included the following discussion to page 8, lines 3-5: "Since 4F11 binds to site Ø and is

prefusion-specific, 4F11 likely neutralizes HMPV by preventing the conformational switch of F from a prefusion to postfusion state, a step essential for viral entry.”

- b. We have also added a new **Fig. S5 and S6** comparing the binding of 4F11 versus M8C10 with the HMPV F trimer. We describe these figures on page 4, lines 28-35: “Indeed, when we model the trimer with two or three Fabs of 4F11 bound, the Fabs clash with each other, suggesting that there is structurally only enough room for binding by a single 4F11 Fab at a time (**Fig. S5**). M8C10 is another mAb that causes trimer dissociation, but it binds an epitope that is buried deep within the trimerization interface. M8C10 interferes with trimer formation likely by clashing with the non-binding protomers of the trimer (**Fig. S6**). In contrast, 4F11 likely facilitates trimer dissociation when more than one Fab tries to bind the trimer, because the Fabs clash with each other (**Fig. S6**).” We have also included more in-depth discussion on the potential mechanism of neutralization by 4F11 and how this compares to M8C10 on page 8, lines 7-17: “The HMPV prefusion F-specific monoclonal antibody M8C10 also disrupts trimer formation, similar to 4F11 (Xiao, et.al., *J Virol*, 2023). However, unlike 4F11 which approaches vertically to bind a quaternary epitope at the apex, the M8C10 epitope is buried deep within the core of the trimer, near antigenic sites II and III, that may only be accessible when the trimer is widely open during the process of “breathing”. Compared to 4F11 (IC₅₀ range 1.0 to 10.7 ng/mL), M8C10 has relatively weaker neutralization potency (IC₅₀ range 371 to 661 ng/mL). Neutralizing antibodies targeting the trimer interface of type 1 fusion proteins, like M8C10 for HMPV and several anti-HA antibodies for influenza, neutralize virus by disrupting the trimer thereby preventing subsequent fusion and viral entry (Bangaru, et. al., *Cell*, 2019; Turner, et. al., *PLoS Biol*, 2019; Zost, et. al., *J Clin Invest*, 2021). Therefore, it is possible that trimer dissociation by 4F11 may be another mechanism that contributes to in vitro neutralization and in vivo efficacy.”
2. The authors suggest that the primary goal is to develop an antibody for therapeutic indication against hMPV. In that case, the role of Fc should be discussed and evaluated in in vitro (such as Fc mediated assays) and/or in vivo studies. The authors performed in vivo experiments to demonstrate the protection efficacy of these antibodies as a therapy. While the data did suggest that these antibodies could offer protection, the lack of benchmark antibodies in the in vivo study prevents comparison of these antibodies to others previously described in the field.
 - a. Please see our response above to Reviewer #1, comment #6 regarding the efficacy of monoclonal antibodies as therapy for respiratory viral infections.

- b. We have also included a more in-depth discussion of the role of Fc effector functions in the context of therapy on page 10, lines 1-10: Notably, similar to 4F11, sotrovimab and REGN-COV2 were formulated as IgG1 with preserved Fc effector function. The relative contribution of Fc effector function compared to neutralization for protection against disease in the setting of treatment may be dose-dependent (DiLillo, et. al., *Nat Med*, 2014; Yamin, et. al., *Nature*, 2021). For instance, in a hamster challenge model of SARS-CoV-2 infection, a clinically relevant dose of a Fc receptor binding null variant of a potent neutralizing antibody had equal potency to wild-type or Fc receptor binding enhanced variants, indicating that Fc independent neutralization was the primary mode of protection. However, at lower doses, Fc effector functions contributed to decreased viral replication and decreased weight loss. Future studies in hamsters will be needed to define the relative contribution of Fc effector functions to protection at varying doses of 4F11.”
 - c. Please see our response above to Reviewer #1, comment #5 regarding the addition of a benchmark antibody to the in vivo experiments.
3. Specific Comments: P1 line 21 and 31: should not use the word “broad” or “broadly” here and throughout. hMPV F is highly conserved between the two subtypes, it is expected anti-F antibodies cross-react to both subtypes and different strains. In fact, this is true for most previously described hMPV F antibodies.
 - a. We thank the reviewer for this feedback and have removed the term “broad” and “broadly” as suggested.
4. P1 line 40-43: should indicate disease burden is also high in older adults. Also, should clarify the approved RSV vaccines are for older adults, while there are two approved RSV monoclonal antibodies indicated for infants.
 - a. We thank the reviewer for this feedback and included the following information on page 1, lines 42-43: “Disease burden is also high for adults aged 60 years or older (Kulkarni, et. al., *Lancet Healthy Longev*, 2025; Walsh, et. al., *Arch Intern Med*, 2008).
 - b. We have modified the sentence on page 1, lines 43-45 to clarify that RSV vaccines are approved for older adults: “While several vaccines for older adults to prevent infection caused by respiratory syncytial virus (RSV), a virus related to HMPV, were approved by the Food and Drug Administration in 2023, protective vaccines for HMPV are not yet clinically available.
 - c. We have also modified the sentence on page 7, lines 9-12 to clarify that RSV monoclonal antibodies are approved for prophylaxis in infants: “For comparison, RSV antibodies such as palivizumab and nirsevimab are

clinically administered for prophylaxis in infants at doses higher than that used in the present study (15 mg/kg and approximately 10-20 mg/kg, respectively) (Boukhvalova, et. al., *Antivir Chem Chemother*, 2018; Domachowske, et. al., *Pediatrics*, 2024; Clegg, et. al., *J Clin Pharmacol*, 2024)

5. P2 line 40: “subtypes” instead of “strains”.
 - a. We have made this correction and changed the terminology to “subtypes”.

6. P2 line 45: I am not convinced the construct (GCN4) used in Wen et al. is stabilized in prefusion conformation. Authors should discuss rationale for this antigen choice in the B cell sorting experiment.
 - a. We agree with the reviewer that, of all the recombinant versions of HMPV F used in this paper, recombinant HMPV F with GCN4 alone is relatively less stable. However, the original paper describing this construct did observe a mixture of prefusion and postfusion conformations of the GCN4-HMPV F by negative stain electron microscopy (Wen, et. al., *Nat Struct Mol Biol*, 2012). We used GCN4-HMPV F as our probe for the B cell sorting experiments, because, our initial experiments were performed in 2021, and this was the only recombinant version of HMPV described at the time that demonstrated a prefusion conformation by structural analysis. Later in 2022, DS-CavE2 was reported (Hsieh, et. al., *Nat Commun*, 2022). It was serendipitous that we performed our B cell sorting experiments with GCN4-HMPV F, even though this was relatively less stable than the newer recombinant versions of HMPV F, because GCN4-HMPV F lacked the proline mutation at amino acid 185. We have clarified this point on page 8, lines 21-31: “We successfully identified 4F11 in our antibody discovery campaign because we had serendipitously used HMPV F without the Pro185 stabilizing mutation. GCN4-stabilized HMPV F was first described in 2012 and newer, more stable constructs of HMPV F have since been reported, including DS-CavES2 in 2022, v3B in 2023, and MPV-2c in 2024 (Wen, et. al., *Nat Struct Mol Biol*, 2012; Hsieh, et. al., *Nat Commun*, 2022; Ou, et.al., *PLoS Pathog*, 2023; Bakkers, et. al., *Nat Commun*, 2024). We used GCN4-stabilized HMPV as the probe in our B cell sorting experiments, because this was the only recombinant version of HMPV F reported at the time to adopt a prefusion conformation. Structural analysis of GCN4-HMPV F has revealed that a mixture of prefusion and postfusion conformations is present, indicating that GCN4-HMPV F is relatively less stable compared to newer versions of recombinant HMPV F (Wen, et. al., *Nat Struct Mol Biol*,

2012). However, if newer constructs with the Pro185 mutation had been used to sort B cells, then 4F11-like antibodies would have been missed.”

7. P3 line 10-11: surprised to see only 6 out of 1140 clones neutralize hMPV A2. How many of them neutralize B strain? How many of them bind preF from A strain? Authors should discuss whether the majority of the B cells binds preF but couldn't neutralize hMPV.
 - a. The reviewer raises a good question about the overall frequency of B cells that bind HMPV preF, neutralize HMPV A subtypes, and neutralize HMPV B subtypes. We focused on functionally screening supernatant for neutralization of HMPV A2, since our objective was to identify highly potent antibodies that could neutralize HMPV subtypes A and B. As a result, in our screen, we did not determine the frequency of B cells producing antibodies that could bind preF and that could neutralize HMPV B subtypes. To further contextualize the frequency of neutralizing B cells in our screen with the published literature on the frequency of B cells in the human repertoire that bind HMPV preF and neutralize HMPV, we have added the following discussion to page 7, lines 21-37: “Others have observed binding to HMPV preF by 0.07-3.69% of class-switched B cells (Rush, et. al., *Cell Reports*, 2022; Rappazo, et. al., *Immunity*, 2022). The expressed antibodies from 22-69% of these B cells had detectable neutralization of HMPV at concentrations up to 10 µg/mL. The majority of HMPV-neutralizing B cells (75%) were able to produce antibodies that neutralized both subtypes A and B (Rush, et. al., *Cell Reports*, 2022). In comparison, our study identified a relatively lower frequency (0.5%) of neutralizing B cells among isotype-switched HMPV preF-binding B cells, because our screening strategy was designed to functionally select for B cells that produce highly potent neutralizing antibodies upfront, without the need to sequence, clone, and produce monoclonal antibodies from all HMPV F-binding B cells. This screening strategy leads to detectable IgG in the supernatant of approximately 40-65% of wells with individually sorted B cells (Whaley, et. al., *J Immunol Methods*, 2020; Boonyaratanakornkit, et. al., *Nat Commun*, 2023). Of the wells with detectable IgG, approximately 35-100% have antibodies in culture supernatant that bind to the same antigen used as the B cell probe. The median IgG level is approximately 4 ng/mL, although some B cells can produce over 475 ng/mL (Whaley, et. al., *J Immunol Methods*, 2020). Therefore, this screening strategy favors the selection of neutralizing antibodies with an IC₅₀ in the ng/mL range, like 4F11, and tends to miss weaker neutralizing antibodies. Furthermore, the recombinant GCN4-HMPV F used as the B cell probe is known to contain a mixture of

prefusion and postfusion conformations of HMPV F (Wen, et. al., *Nat Struct Mol Biol*, 2012). Therefore, HMPV postF-binding B cells were also likely individually sorted into wells and contributed to the relatively lower frequency of neutralizing B cells observed in our screen.”

8. P5 line 38: is there a structural explanation as to how N466K could compensate K179E?
 - a. The reviewer raises a good question about the mechanism of epistasis for the N466K mutation. Although we do not currently have a structural explanation, we have added the following discussion to page 9, lines 9-14: “A secondary mutation N466K appears to have an epistatic effect by partially compensating for the fitness cost associated with K179E. These two amino acids, however, are on opposite ends of the trimer and do not interact in either the prefusion or postfusion conformation. Further biophysical and structural characterization of F with K179E alone and in combination with N466K are needed to elucidate the mechanism of attenuation and partial compensation.”

9. P6 line 17 and 21: I believe the correct figure cited should be Fig 4, not Fig 5.
 - a. The figure numbers have been corrected.

10. P6 line 24-27: the dose comparison with the cited previous studies is apple to orange, as they were for prophylaxis. Especially the last sentence on the clinical dose of palivizumab and nirsevimab is not relevant here.
 - a. We agree with the reviewer that a direct comparison of potency between prophylactic and therapeutic dosing of monoclonal antibodies cannot be made. Our intent was to indicate that the doses used in our in vivo experiments are lower than the doses of other monoclonal antibodies in clinical use, which is an important consideration from the standpoint of toxicity with escalating doses. We have clarified this comparison by deleting the sentence, “In contrast, we found in a previous study that hamsters that received palivizumab at a higher dose of 5 mg/kg as prophylaxis experienced breakthrough lung infection and had no reduction in viral replication in nasal turbinates,” and adding the following sentence to page 7, lines 12-14: “Therefore, the dosage used in our in vivo experiments is lower than the dosages of other monoclonal antibodies in clinical use, which is an important consideration from the standpoint of potential toxicity in humans.”

11. P6 line 36: please clarify the “1:1” stoichiometry is Ab to preF trimer.

- a. We have clarified this point and reworded the phrase to specifically state: “1:1 stoichiometry of 4F11 Fab : preF trimer”.
12. P21 line 16: “WT indicates the same WT hMPV A2 expressing a GFP reporter” – it is confusing that “WT” actually refers to a GFP reporter virus, suggest to rephrase.
- a. We have modified the text in the legend for **Fig. 1** to specifically state: “WT-GFP indicates the same HMPV-A2 expressing a GFP reporter used in the neutralization assay shown in (c). This recombinant virus was generated based on the wild-type sequence of CAN97-83.” We have also modified the label in the legend to read, “WT-GFP”.
13. Reviewer #2 (Remarks on code availability): I can't access the website provided: <https://github.com/jbloombab/Virus-Antibody-Passaging/tree/main>
- a. The link is now active and accessible.

Reviewer #3 (Remarks to the Author):

Reviewer #4 (Remarks to the Author):

1. The use of 1:1 stoichiometry vs 3:1 stoichiometry terminology used throughout the paper is not correct. hMPV F is a homotrimer, so the majority of antibodies discovered previously are actually 1:1 (1 Fab per 1 F monomer), and it is 3:1 in this case.
 - a. The reviewer is correct that the majority of antibodies previously discovered bind with 1 Fab per 1 F monomer, or 3 Fabs per 1 preF trimer. In contrast, 4F11 binds with 1 Fab per 1 preF trimer. We have clarified that the stoichiometry is in reference to the preF trimer and now specifically state on page 1, lines 26-28: “4F11 targets an epitope located at the apex of the prefusion F protein (site Ø) with a 1:1 stoichiometry of Fab to trimer, distinct from the 3:1 stoichiometry observed with other HMPV site Ø antibodies.”
2. Is it possible that two or three Fabs could bind? Provide a size exclusion curve comparing this complex to that of one where other previously discovered Fabs binds in a 1:1 ratio.

- a. Based on our structural analysis, this is not possible since a second or third Fab would clash with the first Fab when they try to bind the trimer. We have included a new **Fig. S5** showing this clash, and we have added the following description to page 4, lines 28-31: “Indeed, when we model the trimer with two or three Fabs of 4F11 bound, the Fabs clash with each other. This suggests that there is structurally only enough room for binding by one 4F11 Fab at a time to the trimer (**Fig. S5**).” Furthermore, in our cryo-EM analysis, we did not observe any particles with 2-3 Fabs bound to trimer.
3. Given the very high sequence conservation between hMPV A and B genotypes, the bait and switch methodology appears to be overstated. Based on the hundreds of antibodies previously isolated for hMPV F, it appears most neutralize both genotypes anyways.
 - a. The reviewer is correct that most HMPV-neutralizing antibodies neutralize both subtypes A and B of HMPV. However, previous studies of the isotype-switched B cell repertoire have observed that approximately, 25% of HMPV-neutralizing B cells are subtype-specific (Rush, et. al., *Cell Reports*, 2022). We have clarified this on page 2, lines 39-43: “Although the majority of HMPV-neutralizing B cells (approximately 75%) produce antibodies that neutralize both subtypes A and B (Rush, et. al., *Cell Reports*, 2022), we leveraged a ‘bait and switch’ strategy to further enrich for B cells that could neutralize both strains. This strategy is based on the rationale that B cells capable of binding to one strain, while neutralizing another strain, are more likely to cross-neutralize both strains.”
4. Is the original isotype of the sequenced B cells that produced the monoclonal antibodies known?
 - a. Based on sequencing, the isotype of monoclonal antibodies 4E11 and 4F11 is IgG. Sequencing analysis was unable to determine the isotype of 3B5. We have added this information to page 3, lines 14-16: “Based on sequencing analysis, 4E11 and 4F11 were derived from B cells expressing the IgG isotype. Our sequencing analysis was unable to determine the isotype for 3B5.”
5. Page 4 line 1: The hMPV F protein is known to dissociate into monomers easily, it is not appropriate to say that 4F11 “caused” the trimer to dissociate. However, please provide the data for this in the supplemental section.
 - a. We agree with the reviewer that HMPV F is relatively unstable and known to dissociate on its own into monomers over time. However, the mechanism of binding by 4F11 to the trimer also likely facilitates trimer dissociation.

Please see our response “b” to Reviewer #2, comment #1 regarding a comparison to other neutralizing antibodies that facilitate trimer dissociation.

- b. We have also included a new **Fig. S2** comparing the size exclusion chromatography tracings of the DS-CavEs2 P185A trimer in the presence and absence of 4F11 to show that 4F11 facilitates trimer dissociation in vitro. We discuss this new figure on page 4, lines 5-12: “In contrast to 4F11, MxR binds across protomers and could help stabilize the trimer. The addition of MxR at a 3:1 molar ratio of Fab : DS-CavEs2 P185A trimer led to a peak in the SEC representing intact trimer bound to three Fabs (**Fig. S2**). To help stabilize the trimer, MxR Fab was precomplexed with DS-CavEs2 P185A trimer before the addition of 4F11 Fab. After addition of 4F11 to this complex, the peak representing the trimer prebound to three MxR Fabs disappeared and a new peak appeared that was shifted to the right and exited the SEC at a fraction corresponding to the size of the F monomer bound to two Fabs (one Fab each of MxR and 4F11) (**Fig. S2**).”
6. The resolution of the cryoEM structure is low, which limits the significance of the work. Perhaps trying to complex with MPE8 and 4F11 would help improve the resolution, since MPE8 binds across protomers. Alternatively, complexing 4F11 with DsCavEs2-IPDS could work since this construct has interprotomer disulfide bonds.

 - a. We are grateful for the reviewer’s feedback. We did attempt to stabilize the trimer by adding the monoclonal antibody MxR, which is a site III-binding monoclonal antibody, similar to MPE8, and that also binds across protomers. However, we still only observed monomers. Please see our response “b” above to comment #5.
 - b. Although we did not specifically complex 4F11 with DsCavEs2-IPDS, we did use v3B which is a version of HMPV preF that is stabilized with the same interprotomer disulfide bonds (V84C-A249C) as DsCavEs2-IPDS. We have added this clarification to page 4, lines 17-19: “Recombinant v3B is a prefusion stabilized variant of HMPV F that includes amino acid substitutions V84C-A249C to generate interprotomer disulfide bonds, similar to another variant DSCavEs2-IPDS (Ou, et. al., *PLoS Pathog*, 2023; Banerjee, et. al., *Proc Natl Acad Sci U S A*, 2022).”
7. A map to model image of the whole complex and interaction site should be provided given the low resolution.

 - a. We have added a new **Fig. S4** showing the map-to-model fit at the interface.

8. Does 4F11 bind other viral glycoproteins similar to antibodies like 2G12, given its interaction with glycan residues? At a minimum, provide binding data with pre-fusion RSV F or some other negative control antigen.
 - a. Unlike the 2G12 monoclonal antibody which specifically binds only the high mannose patch of HIV gp120 with little to no contact with the polypeptide, 4F11 binds mostly to amino acids at the apex of HMPV preF. Since contact by 4F11 with the glycan at Asn172 contributes only a fraction of the overall buried surface area, we would not expect 4F11 to bind non-specifically to other viral glycoproteins. In addition, 4F11 is not expected to bind another viral glycoprotein like RSV F, because our B cell sorting strategy selected for B cells that could bind HMPV but not RSV F (**Fig. 1a, b**). To further demonstrate this, we have added a new **Fig. S10b** and discuss these binding results on page 5, lines 8-12: “Since contact by 4F11 with the glycan at Asn172 contributes only a fraction of the overall buried surface area, we would not expect 4F11 to bind non-specifically to other viral glycoproteins. Indeed, 4F11 does not bind to RSV preF (**Fig. S10b**), as expected since our B cell sorting strategy already selected for B cells that could bind HMPV but not RSV preF (**Fig. 1a, b**).”

9. Is the glycan important for binding? N172T is used, but why was threonine chosen and not aspartate?
 - a. We removed the glycan three different ways: 1) N172T substitution in the glycan motif; 2) T174A substitution in the glycan motif; and 3) EndoH treatment of recombinant HMPV F after production in GnTI⁻ 293 cells. We have added the results of EndoH treated HMPV produced in GnTI⁻ 293 cells as new **Fig. S10a**. These 3 lines of evidence, along with the structural data, together indicate that the glycan is important for binding by 4F11.
 - b. The reviewer brings up an excellent point about the rationale for mutating the asparagine to threonine. We have added a new **Fig. S13** and the following explanation to page 6, lines 31-33: “The N172T substitution was chosen, because we found a single HMPV F sequence with a threonine at position 172 in Nextstrain. All other 998 sequences (99.9%) in Nextstrain had an asparagine.” We have also added a discussion of the potential effect of this mutation on viral escape and fitness to page 9, lines 22-27: “Mutation of the glycosylation motif at Asn172 represents another potential route of escape. However, only 1 out of 999 (0.1%) sequences on Nextstrain have a N172T substitution. This rarity in circulating HMPV strains likely reflects a substantial fitness cost of the mutation. Indeed, site-directed mutagenesis and reverse genetics were used to recover recombinant HMPV lacking

glycosylation at position 172, and this virus was profoundly attenuated in vitro and in vivo (Zhang, et. al., *J Gen Virol*, 2011).”

10. Plaque purification of viral clones is mentioned in the results, but the methods don't mention picking individual plaques, how was this done?
 - a. We have added a description of the plaque purification technique to page 16, lines 37-43: “Vero cells were plated in 24-well plates and inoculated with serial 1:10 dilutions of virus, overlaid with methylcellulose containing 1.2% trypsin (Gibco, cat#25300054), and cultured for 5-7 days. Individual plaques were identified with fluorescent microscopy and picked with a P200 pipette tip. The aspirate containing the plaque was transferred onto Vero cells in 96-well plates containing 0.234 µg/mL of 4F11. After 7 days, supernatant was transferred to Vero cells in a T25 flask containing 0.234 µg/mL of 4F11. Purified virus was re-sequenced by Sanger sequencing.”

11. There are no cells in the last gated population in Figure 1 b.
 - a. The last flow plot is derived from cells contained in the flowthrough fraction after magnetic enrichment. We included the flow plot to demonstrate the effectiveness of our magnetic enrichment strategy, which leads to depletion of antigen-specific cells in the flowthrough fraction. Therefore, the lack of cells is expected and further evidence of successful enrichment. We have clarified this in the figure legend by adding the following description: “The flowthrough fraction contains cells that did not bind the magnet and were thus depleted of antigen-specific cells.”

12. In supplemental 4, provide the raw data as well as the fit curves on top in the figures.
 - a. Please see our response to reviewer #1, comment #3. We have added the fitted curves to **Fig. S8**.

REVIEWERS' COMMENTS

Reviewer #1 (Remarks to the Author):

1. Thank you for addressing our three major concerns from the previous review (reviewer #1). However, after careful consideration of the revised manuscript and the authors' responses, we still have reservations about all three major concerns that require further attention.

For Major concerns 1 and 2

We appreciated authors made efforts to show the model-to-fit interface in FigS4 and figured out the heavy and light chains should flip. However, cryo-EM map seems not meet the high quality for the high-impact factor journal and dataset seems suffers from orientation bias. I will encourage the authors collect another cryo-EM dataset to achieve higher resolution, which enables accurate model building. If glycan N172 plays a key role in contacting with 4F11 Fab, then authors should also show the map and model at this binding interface instead of only showing glycan alone on F.

- a. We agree that our cryo-EM data set would ideally have higher resolution. We have made numerous cryo-EM grids with various conditions, including pre-complexing with other monoclonal antibodies, trying different prefusion-stabilized versions of HMPV F, and trying different detergents, but we still have observed an orientation bias. We have submitted the best dataset available to enable modeling the antibody interface. We think that it is reasonable to expect that higher-resolution analysis would overall reveal the same binding mode. We have also modified the following sentence in the discussion on page 9, lines 8-11: "Further biophysical and higher resolution structural characterization of 4F11 in complex with F, F with K179E alone, and F with K179E in combination with N466K are needed to elucidate the mechanisms of neutralization escape, growth attenuation, and partial compensation." As suggested, we have also now added **Supplemental Figure 4c** to show the map around the glycan at the binding interface.
2. For major concern 3
We appreciated authors added fit and experimental curves for BLI. However, KD value of 4F11 Fab bound to monomer in the table seems incorrect if KD is truly derived from k_{on} and k_{off} .
 - a. The reviewer is correct to point out that the K_D value of 1.01E-07 differs from the ratio $\frac{k_{dis}}{k_{on}} = 0.74E-07$ for the monomer in **Figure S8**. The reason for this slight

discrepancy is because the values reported in **Figure S8** include the average K_D , average k_{dis} , and average k_{on} of each binding curve. Since K_D is a ratio equal to $\frac{k_{dis}}{k_{on}}$, the average K_D can be calculated either by 1) averaging the K_D of each curve; or 2) taking the ratio $\frac{\text{average}(k_{dis})}{\text{average}(k_{on})}$. The first approach is an “average of ratios”, which weights each individual ratio ($\frac{k_{dis}}{k_{on}}$) equally. The second approach is a “ratio of averages”, which weights each group (k_{dis} and k_{on}) equally. Mathematically, the “average of ratios” does not always equal the “ratio of averages”. In general, taking the “average of ratios” is preferred and more robust, because this approach gives each experimental condition (each binding curve) equal weight and, as a result, provides a better representation of the central tendency for affinity across multiple experiments (Liu, et al., *Mabs*, 2014; Kamat, et al., *Anal Biochem*, 2017; Lee, et al., *J Mol Biol*, 2020). In contrast, taking the “ratio of averages” is less preferred, because k_{dis} and k_{on} are derived from fitting kinetic data to non-linear models. The raw data for K_D , k_{dis} , and k_{on} can be found in the file “Source data.xlsx”. To avoid confusion, we have clarified in the figure legend for **Figure S8** that the values reported are the average K_D , average k_{dis} , and average k_{on} of each binding curve.

Reviewer #2 (Remarks to the Author):

The revised manuscript has addressed my comments.

Reviewer #3 (Remarks to the Author):

Reviewer #4 (Remarks to the Author):

The authors addressed all of my concerns.